**DOI: 10.1038/ncomms15604**　　**OPEN**

# A robot for high yield electrophysiology and morphology of single neurons *in vivo*

Lu Li[1], Benjamin Ouellette[1,*], William A. Stoy[2,*], Emma J. Garren[1], Tanya L. Daigle[1], Craig R. Forest[2,3], Christof Koch[1] & Hongkui Zeng[1]

Single-cell characterization and perturbation of neurons provides knowledge critical to addressing fundamental neuroscience questions including the structure–function relationship and neuronal cell-type classification. Here we report a robot for efficiently performing *in vivo* single-cell experiments in deep brain tissues optically difficult to access. This robot automates blind (non-visually guided) single-cell electroporation (SCE) and extracellular electrophysiology, and can be used to characterize neuronal morphological and physiological properties of, and/or manipulate genetic/chemical contents via delivering extraneous materials (for example, genes) into single neurons *in vivo*. Tested in the mouse brain, our robot successfully reveals the full morphology of single-infragranular neurons recorded in multiple neocortical regions, as well as deep brain structures such as hippocampal CA3, with high efficiency. Our robot thus can greatly facilitate the study of *in vivo* full morphology and electrophysiology of single neurons in the brain.

[1] Allen Institute for Brain Science, 615 Westlake Avenue N, Seattle, Washington 98109, USA. [2] Wallace H. Coulter Department of Biomedical Engineering, Atlanta, Georgia 30332, USA. [3] George W. Woodruff School of Mechanical Engineering, Georgia Institute of Technology, Atlanta, Georgia 30332, USA. * These authors contributed equally to this work. Correspondence and requests for materials should be addressed to L.L. (email: lul@alleninstitute.org).

The brain processes information through intricately interconnected neurons. To understand how the brain guides behaviour, it is necessary to characterize and perturb neurons *in vivo*, in which the underlying neurobiological substrates are best preserved. Ideally such characterization and perturbation should have single-cell resolution, because neurons demonstrate great diversity in their morphology, physiology, connectivity and genetics[1–4]. A major advance towards this goal is the invention of all-optical methods[5,6] to record and manipulate activity of single neurons *in vivo*, but it requires genetically modified animals and its application is restricted to the sheath of tissue accessible to two-photon microscopy. Another issue with optical methods is the lack of direct physical access, making it difficult to label, manipulate or harvest chemical/genetic contents of recorded neurons. Labelling the neurons investigated *in vivo* allows obtaining and correlating multiple modalities of data including full morphology, function/physiology and/or genetics at the single-neuron level, which is critical to addressing long-standing neuroscience questions such as the structure–function relationship and neuronal cell-type classification (for example, the Correspondence problem)[7]. Neuronal full morphology is a pivot piece of data, because it not only delineates the range and pattern of neurons' input and output, but also provides the anchor linking function, connectivity and genetics together. A representative case is the long-range projecting neurons, which consist of ∼80% of the entire neuronal population in neocortex[8,9] and extend their axons far away from the soma to connect distal brain regions[10,11]. Efforts have been made to characterize them in reduced preparations (for example, brain slices)[12–15], but acquired information is fragmental due to the fact that a great portion of neurites are truncated during tissue processing, which results in permanent information loss. Within brain slices, only local dendrites and limited segments of axons are preserved, and the network-dependent neural responses are completely absent. Thus *in vivo* single-cell experiments with physical probes are necessary to reveal and correlate the full morphology with functional properties.

To establish the aforementioned correlation, single neurons need to be recorded and labelled at a large scale. A robotic system like the Autopatcher and similar tools[16,17] is thus desirable. These automated systems simplify the heuristics of manual patch-clamp electrophysiology to an algorithm with a defined series of steps for localizing the pipette to a cell of interest, gigasealing and breaking-in, thereby greatly facilitate electrophysiological research[16,17]. But these automatic platforms are not designed for efficiently labelling neurons *in vivo* for full morphology reconstruction. So far, labelling recorded neurons *in vivo* for their full morphology has been done exclusively manually, which is low-yield and requires high skills. For example, microiontophoresis of biocytin or its derivatives with micropipettes has been considered the gold-standard technique due to its great success in labelling brain cells *in vivo*[18,19], but because it requires a long filling time to ensure sufficient amount of materials entering the cell, in practice high-quality single-cell labelling could be difficult to achieve due to contamination. Moreover, biocytin diffuses to reach distal processes of the cells, which weakens signal so the completeness of *in vivo* labelling could be an open question. In addition, this method is technically difficult and requires significant expertise and training. Two-photon guided *in vivo* single-cell electroporation (SCE) has been introduced[20,21] and recently automated[22]. However, its applications are restricted to the superficial brain regions accessible to two-photon microscopy. Juxtacellular electroporation and whole-cell (blind) recording, on the other hand, have been manually conducted in deep brain structures but are technically demanding[23–28]. Whole-cell experiments also require careful and highly skilled preparation to re-seal the membrane at the end of filling and despite recent efforts, suffer from low yield for delivery of genetic constructs through the patch pipette[29,30]. In summary, *in vivo* single-cell experiments usually require a considerable amount of efforts including experienced laboratory personnel, extensive training and labour, not to mention the low efficiency from which many *in vivo* experiments suffer. Thus a high-efficiency, cost-effective and easy-to-use method is needed.

Here we present a number of high-yield *in vivo* single-cell experiments using the ACE (Automatic single-Cell Experimenter), a robot that automates *in vivo* SCE and blind cell-attached recording to detect, record, and/or manipulate/label single neurons (Fig. 1 and Supplementary Fig. 1, also see Supplementary Movie 1). ACE features a modular design, consisting of commercially available hardware components controlled by customizable, publicly available, LabView-based software (Fig. 1a, Methods section). This design has several advantages. First, automation will improve the yield by performing a set of optimized experimental procedures in a standardized manner, which will minimize the variability during experiment execution and reduce the dependence on experimentalists. Second, by automating *in vivo* SCE, ACE can manipulate the chemical and/or genetic contents of single neurons by delivering a variety of extraneous materials into single cells based on the electric charge (electrophoresis) and/or concentration gradient (passive diffusion) through glass micropipettes, the best available option to gain physical access to neurons in deep brain structures. Thirdly, combined with cell-attached recording, ACE can characterize functional properties of single neurons *in vivo* with high spatiotemporal resolution. In addition, through the relay-based switching mechanism, ACE can switch between SCE and electrophysiology seamlessly to record neural responses, or deliver materials into cells, or both. Fourth, ACE does not require a two-photon microscope, making it cost-effective. Finally, blind patch recording and SCE have been manually conducted in multiple animal models[23–30], meaning that ACE can have broad applications in various species in which transgenic animal models are not readily available.

## Results

**ACE records deep neurons with high efficiency and precision.** To efficiently conduct *in vivo* single-cell experiments, the first task for ACE is to maximize the successful detection of neurons. In previous whole-cell studies automatic neuron detection was achieved by passively monitoring the change in pipette resistance ($R_p$) as the pipette advanced into the brain[16,17]. This passive sensing algorithm successfully triggered detections but is less useful to differentiate false-positives (detection triggered by non-neuronal events such as glial cells, blood vessels or clogged pipettes) or misses (for example, inactive neurons). Identification of silent neurons is important for studies in brain regions where a sparse coding scheme is employed[31]. To solve these problems and achieve higher precision, we exploited the excitability of neuronal cells and implemented a detection algorithm consisting of passive sensing of object proximity through $R_p$ increase and active verification by current-injection-evoked action potentials (spikes, Methods section). To verify this detection algorithm (Fig. 1b), in the first set of experiments we deployed ACE for cell-attached recordings in neurons in primary visual cortex (V1) in Isoflurane anesthetized wild-type C57BL/6J mice ($n = 15$). In these experiments (that is, steps 1, 2 and 4 in Fig. 1c), ACE searched for cells between ≥ 500 µm and 1,500 µm underneath the pial surface (Fig. 2a), focusing on infragranular layers (L) 5 and 6,

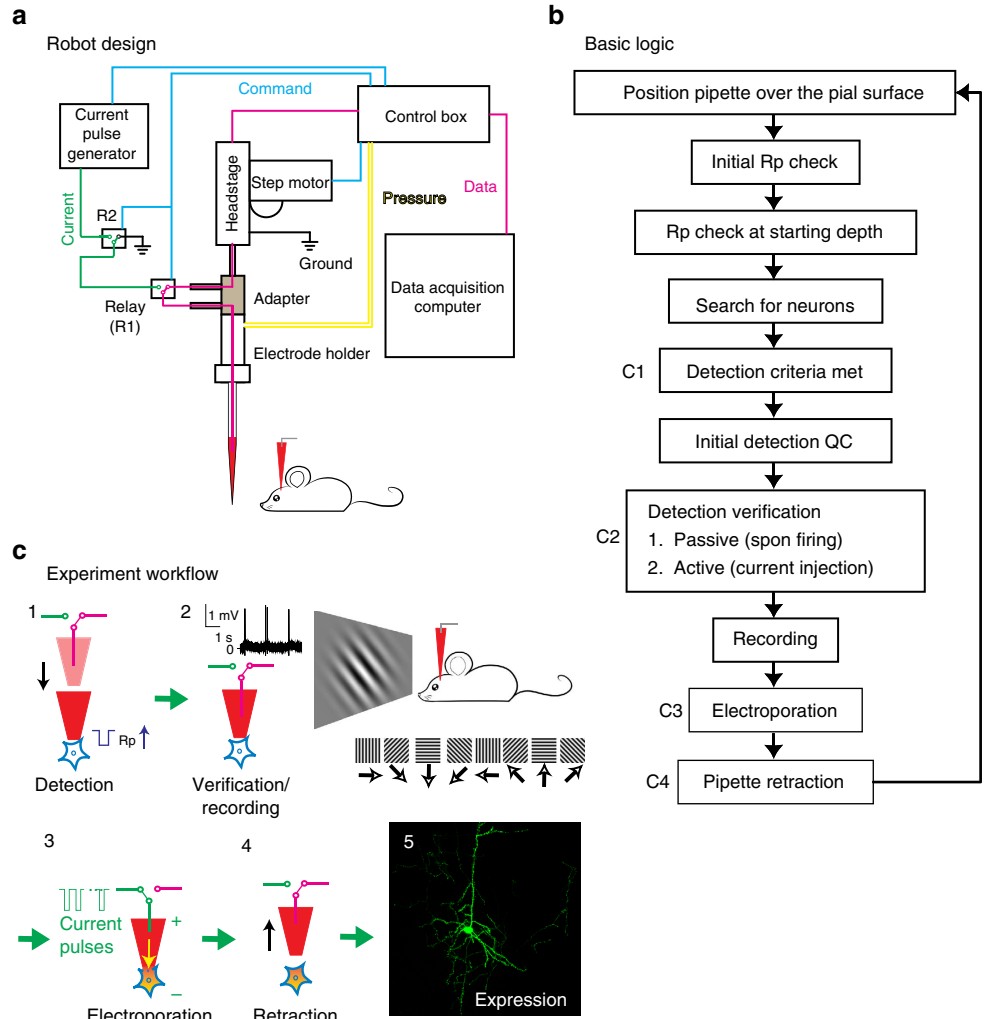

**Figure 1 | Automating *in vivo* single-cell recording and perturbation.** (**a**) Schema of ACE design showing major hardware components of ACE, the Automatic single-Cell Experimenter. The relay-based mechanism allows freely switching between single-cell recording and perturbation (electroporation). (**b**) Block diagram illustrating the basic logic flow of ACE operations for pipette advancement, neuron detection and verification, neural activity recording and electroporation. For detailed information see Methods section. (**c**) Illustration of main steps of representative experiment work-flow for *in vivo* single-cell recording/labelling experiments to study the structure–function relationship in single neurons. These steps are labelled in **b**.

which are difficult to reach by regular two-photon imaging. To better accommodate $R_p$ variations, a relative detection threshold ($R_p$ increase $\geq 8$–10% of original $R_p$), which is similar to previous published studies[17,24], was used. On average $12.5 \pm 0.5$ (mean $\pm$ s.e.m., same below) penetrations were conducted per mouse V1 (9–15 penetrations per animal). Detection events were triggered quickly ($2\,\text{min}:09 \pm 5\,\text{s}$) with high probability (that is, the detection prevalence: $90 \pm 2$%, in total 168/188 penetrations triggered detection, Supplementary Fig. 2A). The depth of detection events was distributed between 500 and 1,000 µm (Supplementary Fig. 2B), consistent with what is known for blind patching[16,17,24]. As in the case shown in Fig. 2b, stable *in vivo* cell-attached recording was achieved by ACE for $\sim 18\,\text{min}$ to characterize visually evoked responses of neurons recorded in V1 before termination by the experimenter (total of nine recordings in five mice). ACE achieved a nearly 90% detection precision (chance of neuron when detection was triggered) in mouse V1 ($87 \pm 4$%, in total 146/168 detections verified as neurons by either spontaneous or current-injection-evoked spiking, Fig. 2c).

Detection verification by current injection significantly increased the hit rate. In a separate set of control experiments without current injections ($n = 11$ mice, 138 penetrations), the detection precision dropped to $61 \pm 5$% in V1 ($P < 0.01$, $t$-test), thus ACE improved the hit rate by reducing 'misses' (false negatives) such as spontaneously inactive neurons, which frequently occur with passive detection algorithm alone. The verification process by current injection (Supplementary Figs 3 and 4) had no significant effect on the average trial time (injection versus no injection: $5\,\text{min}:29 \pm 12\,\text{s}$, 168 penetrations versus $5:39 \pm 16$, 138 penetrations, $P = 0.36$, $t$-test, Fig. 2d). The high precision of detection was consistent among most of the 15 animals (Supplementary Fig. 2A) and showed little dependence on $R_p$, laminar locations or types of pipette solution (saline, K-gluconate based internal solution or artificial cerebrospinal fluid (ACSF)). Furthermore, consistent results were found in deep layers of other cortical regions including a motor area—the primary motor cortex (M1, detection time: $2\,\text{min}:16 \pm 14\,\text{s}$, 8 penetrations in 2 mice) and a frontal area—the anterior lateral motor area (ALM, detection time: $2\,\text{min}:31 \pm 11\,\text{s}$, 84 penetrations in 10 mice), indicating the robustness of our detection algorithm across brain regions (Supplementary Fig. 2C–E). We conclude that with our detection algorithm ACE achieved a high detection and recording efficiency of single-brain neurons *in vivo*.

**ACE labels single neurons in deep brain regions *in vivo*.** Next, we tested ACE's functionality of single neuron labelling, which is the automated SCE module (Fig. 3a). In this set of experiments, we filled pipettes with K-gluconate based internal solution containing DNA plasmid encoding a tracer fluorescence protein (enhanced green fluorescence protein, EGFP). By doing this, we can label detected neurons to obtain their *in vivo* full morphology. Compared with the biocytin filling, which is a standard technique for morphological reconstruction[18,19], our method is advantageous because firstly, the self-generative

tracer will be continuously synthesized then diffuse or be transported within the cytosol; thus, it will better reveal distal processes; secondly, there is no spill-over confounding effects from DNA plasmid because the tracer is spatially restricted to the target cell and can be amplified by antibody staining.

Electroporation was conducted in infragranular layers of V1, the major cortical output layers sending long-range projecting axons to innervate other brain regions, in Isoflurane anesthetized wild-type C57BL/6J mice ($n = 9$). Due to the demonstrated high detection precision of ~90% (Fig. 2 and Supplementary Fig. 2), here for simplification we programmed ACE to search for neurons then activate the relay-based switching mechanism to perform SCE upon detection without post-detection current injections (that is, steps 1, 3, 4 and 5 in Fig. 1c). Regular patch pipettes ($R_P = 7$–$12\,\Omega$) were used[20–22]. Detection time was similar to that of previous electrophysiology experiments (data not shown). A train of unipolar pulses ($-12\,V$ pulses with 1–2 ms pulse width at 50 Hz for 0.3–2 s)[20–23,32] was delivered through the pipette to electroporate the EGFP plasmid. One week after the electroporation, strong native fluorescence signal was observed in individual neurons (Fig. 3b,d,e). Thus ACE successfully altered the genetic content of single-brain cells. Labelled neurons were located in L5 and 6 of V1, matching with our records. The labelling was very strong and revealed rich morphological detail including dense dendritic spines and axonal boutons (Fig. 3b1–4). At each electroporation site, only one fluorescence-positive neuron, not a cluster of neighbouring neurons, was found, indicating a single-cell resolution of manipulation *in vivo*. Unlike the biocytin labelling (Fig. 3c), in these experiments no 'spill-over' labelling of off-site neurons was observed around the electroporation site, or along the electrode tract (Fig. 3b,d,e). In all cases, no diffusive, non-specific 'halo' of auto-fluorescence was observed, suggesting little or no tissue damage was caused by the parameters used for electroporation (Fig. 3b,d,e). More importantly, ACE successfully revealed the distal axons of single-projection neurons down to striatum and brainstem,

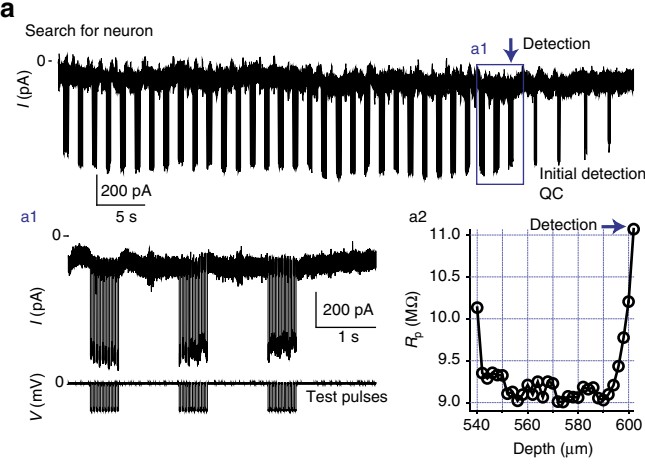

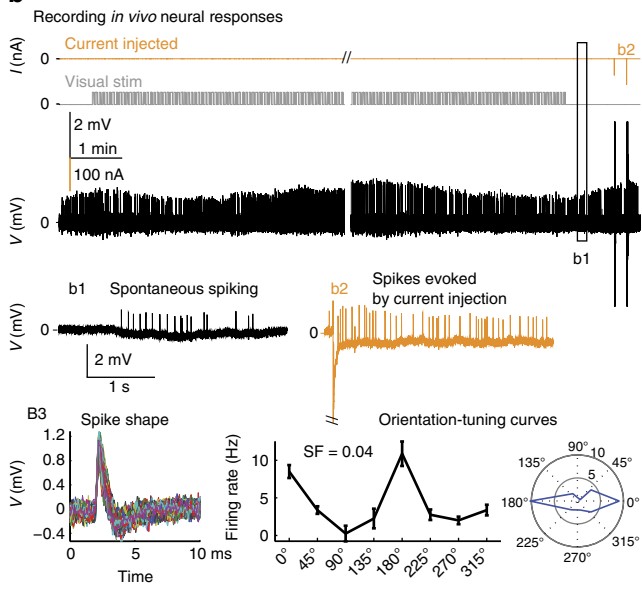

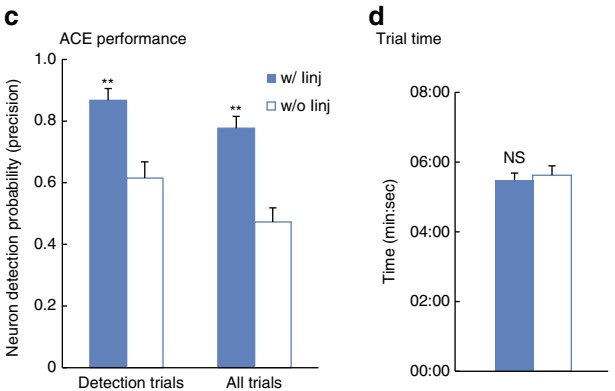

**Figure 2 | ACE detects neurons efficiently with high precision and records neural activity stably in deep brain tissue.** (**a**) Example current trace recorded showing automatic neuron detection by ACE *in vivo*. The pipette was advanced into the brain at $2\,\mu m\,step^{-1}sec^{-1}$. (a1) At each step, a train of ten $-5\,mV$ test pulses was delivered to the pipette and the current responses were measured to calculate the average pipette resistance ($R_P$). $R_P$ was then compared with values calculated at previous steps to see whether the neuron detection criteria would be met. (a2) In this penetration detection occurred within 4–6 μm. A threshold of 10% $R_P$ increase was used. Initial drop of $R_P$ at 540 μm indicated a rejection event (most likely the pipette passed a dendrite or capillary). (**b**) ACE achieves stable recording. The voltage trace (high-pass filtered at 2 Hz) of cell-attached recording of neural responses is shown for the same neuron detected in **a**. Spontaneous activity and visually evoked responses were both recorded. Orientation-tuning curve was mapped in this neuron with eight orientation, five spatial frequencies and one temporal frequency (Methods section). (b1) Normal spontaneous activity after about 18 min of recording indicated no apparent damage to the cytoplasma membrane integrity. (b2) Mild current injection ($-100\,nA$) evoked rapid and reversible increase of firing. Also see Fig. 4 for detection of a spontaneously inactive neuron verified by current injections. (b3) Spike shape (left) and orientation-tuning curves (middle and right) of the same neuron. Only responses to the preferred spatial frequency (0.04 cycle per degree) were shown. (**c**) Our detection–verification algorithm greatly improves the performance of neuron detections. Data were compared for detection trials in which neuron detection criteria were met (that is, the detection precision), and all trials, with or without current injection to verify the detection. (**d**) The verification by current injection had no effect on the average trial time.

spanning across ∼4.0 mm anterio-posteriorly and ∼4.5 mm dorso-ventrally (Fig. 3e,f). With the high quality of labelling, the full morphology of single-projection neurons can be readily visualized, traced and reconstructed (Supplementary Movie 2).

ACE also successfully labelled neurons in brain structures deeper than cortex, for example, hippocampal CA3 (Supplementary Fig. 5). Thus ACE had no apparent laminar or regional preference. ACE could also label individual neurons in the same brain with different plasmids, for example, a plasmid encoding a red fluorescence protein tdTomato, with similar labelling to EGFP (neurons labelled in Fig. 3b and Supplementary Fig. 6D are from the same animal). This allows labelling multiple neurons in the same brain region with well separable processes in following reconstruction, thus the experiment throughput could be increased. Occasionally glial cells were labelled, consistent with the electrophysiology results of low false-positive rate

(Supplementary Fig. 7). Overall 21 neurons, about 16% of all detection events, were recovered by native fluorescence from 130 penetrations in nine mice, comparable to that of published manual electroporation labelling experiments *in vivo*[25,26,28] (Fig. 3g). We also tested saline or ACSF-based pipette solution with ACE. Surprisingly, although detections were successfully triggered, no cells were recovered in either V1 ($n = 13$ mice, 152 penetrations), or M1 ($n = 11$ mice, 88 penetrations) or ALM ($n = 1$ mouse, 10 penetrations) under our conditions, unlike previously published studies in tadpoles[23,32]. One possible reason is the species-dependence of sensitivity to disturbance of cross-membrane ion gradient, thus we only used K-gluconate based internal solution for electroporation experiments.

**ACE facilitates single-cell structure–function correlation.** Scalable research requires high-efficiency neuron labelling. After establishing the feasibility of automatic single-neuron labelling for full morphology reconstruction, we aimed at improving the material delivery efficiency. Assisted with *in vivo* two-photon imaging, we found the regular patching pipettes (7–12 MΩ) we used[22], which are similar to those reported in literature[20–22], showed a high tendency of being clogged after travelling certain distance within the brain tissue *in vivo* but still remain conductive (data not shown). This could be due to the increased viscosity of the DNA-containing solution compared with regular internal solution. This would cause a low material delivering efficiency and thereby reduce the rate of successful labelling. To deal with this issue, we searched for a good balance between successful delivery (which requires a larger tip size) and minimal damage to

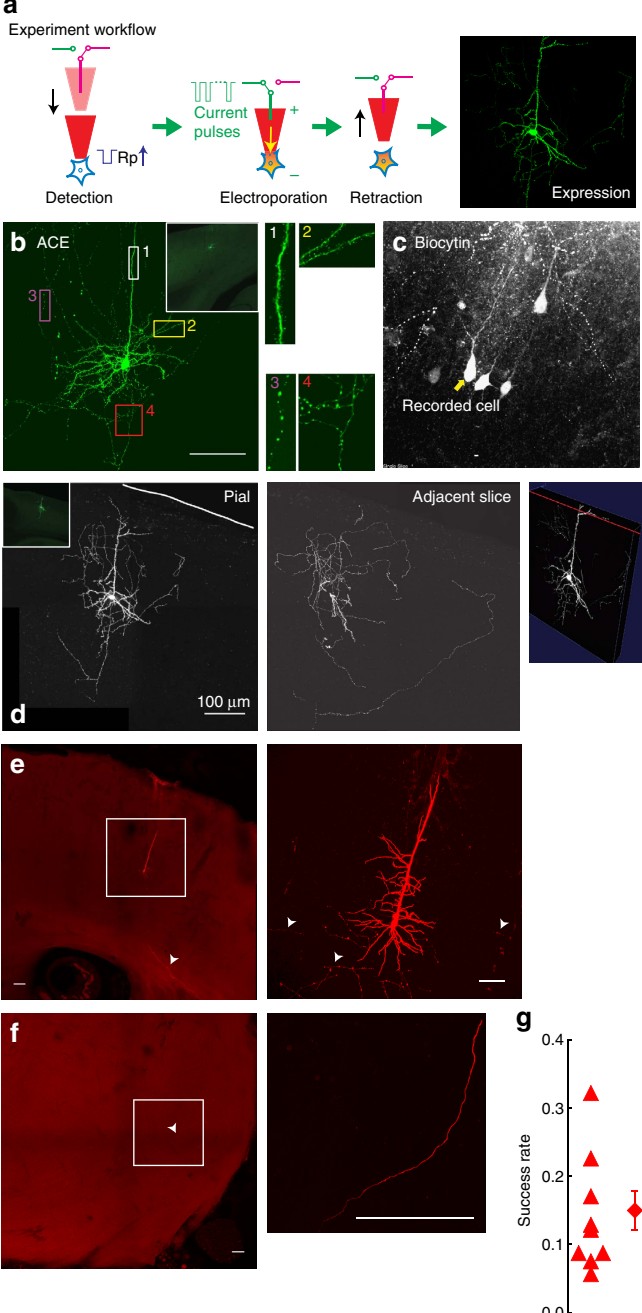

**Figure 3 | ACE reveals the *in vivo* morphology of individual neurons.**
(**a**) Illustration of experiment work flow for *in vivo* single-cell manipulation. (**b**) Z-projection image of a confocal image stack showing the morphology of a V1 L5 neuron electroporated *in vivo* by ACE with the EGFP plasmid. The confocal image stack was taken from a 100 μm-thick coronal section with a 40 × , 1.3 NA oil-immersion objective. Native fluorescence was imaged without antibody amplification. Note the fine detail of dendritic and axonal structure (**b**, 1–4), confirming the high labelling quality. Inset: low magnification epifluorescence image (taken with a 10 × , 0.4 NA dry objective) showing the overview of the soma-containing region in the same coronal section. Note that only this neuron was labelled, and no visible damage or contamination is noticeable. Expression time for this neuron is 7-day. Also see Supplementary Fig. 5D for a tdTomato expressing neuron labelled in the same brain. (**c**) Z-projection image of labelled neurons from a biocytin cell-filling experiment using *in vivo* whole-cell patching. Multiple penetrations were conducted. Coronal sections were cut at 100 μm, processed for Biocytin-Streptavidin reaction then imaged with a 40 × , 1.3 NA oil-immersion objective. Note, classical biocytin cell-filling technique for *in vivo* morphological reconstruction may introduce confounding effects such as spill-over contamination, labelling of multiple cells nearby or along the electrode tract. These issues are not present in experiments with ACE electroporation (**b,d,e**). (**d**) Montages of Z-projection images of another L5 neuron electroporated with the EGFP plasmid. Soma (left) and local processes in adjacent brain sections (middle) are nicely labelled and can be visualized in 3D (right). Native fluorescence was imaged with a 40 × , 1.3 NA oil-immersion objective. (**e,f**). ACE successfully visualizes distal axons millimetres away from the soma thus the full morphology can be readily reconstructed. Montages show local processes (**e**) and distally projecting axon (**f**) of a L5 neuron electroporated with CAG-TdTomato plasmid in M1 *in vivo*. A 10 × , 0.4 NA dry objective was used to take image stacks for left figure panels of **e** and **f**. (Right panels) show the regions within the white boxes in **e** and **f**, respectively. In this neuron the axon can be traced ∼4.0 mm posteriorly and 4.5 mm ventrally (**f**) away from the fluorescence-positive soma. Arrow heads: projecting axons. (**g**) Experiment yield ($n = 9$ mice) is comparable to manual experiments. Scale bar: 100 μm.

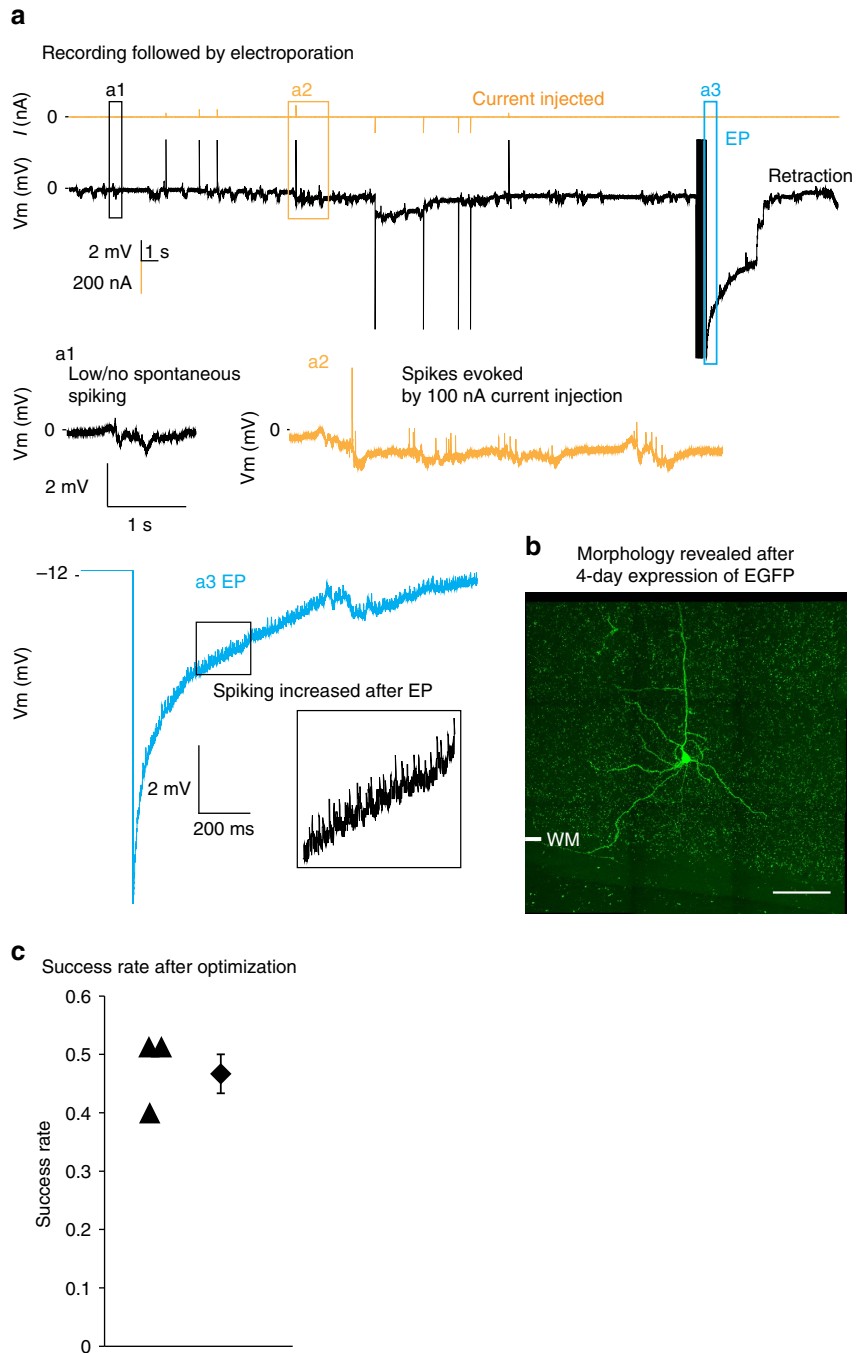

**Figure 4 | Correlating morphology with *in vivo* electrophysiology at single-cell level with high efficiency.** (**a**) Voltage trace of cell-attached recording of a neuron detected at a subpial depth of 768 µm. A threshold of 10% $R_p$ increase was used. (a1) This neuron was spontaneously inactive but voltage fluctuation was observed. (a2) Current injection (100 nA) reversibly evoked spikes. (a3) Electroporation was conducted to deliver the EGFP plasmid into the neuron, increased spiking was observed after electroporation. (**b**) The neuron recorded was recovered in L6 of V1, showing morphological features of pyramidal neurons, which is consistent with the electrophysiological data. Montage of Z-projection images is shown here, note the pyramidal-shape soma and spiny dendrites. Image stacks were taken from the coronal section with a 63 ×, 1.4 NA oil-immersion objective. (**c**) Experiment yield is high across three mice tested. WM, white matter. Scale bar: 100 µm.

the cell (which requires a smaller tip size) by using pipettes with $R_p = 3$–6 MΩ. To prevent post-electroporation cytolysis, electroporation voltage was also lowered so that the transmembrane current is approximately between 1–1.5 µA (ref. 32), which is estimated to achieve the dielectric voltage drop across the cytoplasmic membrane. Intermediate concentration of DNA plasmid (0.1–0.2 µg per µl) was used to improve the delivery efficiency.

ACE then performed standardized blind experiment procedures with optimized parameters combined with active current injection, which resulted in a ∼3-fold increase of success rate (Fig. 4c, compared to Fig. 3g). These experiments completed the entire work flow (Fig. 1c steps 1–5). In the example cell shown in Fig. 4a,b, a low-resistance pipette ($R_p = 4.0$ MΩ) was used (Supplementary Movie 1). This neuron was detected at a relatively deep location (768 µm underneath the pial surface),

was spontaneously inactive and unresponsive to flash stimuli (Fig. 4a1), but responded to current injections with the spiking behaviour of typical pyramidal neurons (average firing rate $\sim 2$ Hz post current injection, instantaneous firing rate up to 60 Hz, Fig. 4a2). Increased spiking (Fig. 4a2) was observed in this cell post-electroporation[25,26,28,33]. After a 4-day expression, we successfully recovered this neuron's pyramidal morphology at matching location and depth. In this animal, electroporation was conducted in 8 of 10 penetrations in which neurons were confirmed after detection. Four neurons were recovered, resulting in a success rate of 50%. We then repeated this standardized process and labelled another six neurons in two more mice. On average the success rate was high ($47 \pm 3$%, $n = 3$ mice) and consistent across three animals (50, 40 and 50%, respectively), higher than most blind manual studies (Fig. 4c)[25,26,28]. Among these cells, we successfully recovered a neuron with $\sim 20$ min of recording in vivo (Supplementary Fig. 8). In our current data set, we did not notice apparent correlation between the length of recording time ($\leq 30$ min) and success rate of labelling.

Electroporation in vivo can label multiple neighbouring neurons if a high voltage is used (Supplementary Figs 5A,6C,D)[23,32]. Although single neurons were recovered at every successful site, due to the high density of pyramidal neurons in cortex, it remains possible that the neuron labelled was one of the targeted neuron's random neighbours, and the targeted neuron failed to survive from the electroporation/labelling. To dismiss this possibility in our experiments, we chose to record and label fast-spiking (FS) inhibitory interneurons ($n = 3$), which are much more spatially sparsely distributed ($\sim 6$% of the entire cortical neuronal population)[4], and physiologically and morphologically distinct from pyramidal neurons. Experiments were conducted blindly in the same way as in Fig. 4. FS cells were identified by their narrow spike-width and high firing frequency ($> 200$ Hz) in response to current injections (Supplementary Fig. 9). Electroporation was then conducted and the labelled neuron was recovered with morphological features including round soma and smooth dendrites, which are typical for FS cells. We thus concluded that ACE could efficiently record and label the same neuron in vivo with real single-cell resolution.

## Discussion

Acquiring and correlating the morphology/connectivity, function/physiology and/or genetics of single neurons will facilitate our understanding of the neuronal structure–function relationship and cell-type classification. Here we present an automated, cost-effective and high-efficiency robot, including the hardware, software and tuned parameter set, to record and manipulate single neurons in vivo. Compared with optics-aided single-neuron recording/labelling[20–22,34], ACE can record and label deep brain neurons that are optically difficult to access. An important target is infragranular projection neurons, which connects distal brain regions together. Recent studies with bulk injections have greatly advanced our understanding of brain connectivity at the mesoscale level[10,35–37], but more data of full morphology are needed at the single-neuron level for these long-range projection neurons. Here we show ACE could record activity and reveal full morphology of single-infragranular projection neurons in mouse brain. This is achieved without any laser or two-photon microscope, making ACE highly cost-effective. Unlike existing manual methods for single-neuron labelling such as whole-cell patch-clamp and juxtacellular labelling[23–30], ACE doesn't require extensive training. And benefited from its operation standardization, ACE obtains results independent of human experimenters' skill level and daily performance change. Its use will thereby, improve

experiment efficiency and consistency. To reach deep brain structures, ACE has to work independently of visual guidance (therefore blindly). But with refined pipette geometry, we demonstrated ACE could recover the morphology of both the excitatory projection neurons and fast-spiking inhibitory interneurons in vivo, even though it is known that blind patch recording is biased towards excitatory neurons[24]. To our knowledge this is the first robot reported that improves the experiment yield and lowers the hurdle of labelling neurons recorded for morphological reconstruction, which has long been considered challenging manually. This system will enable a scalable effort in addressing the question of neuronal structure–function relationship.

Morphology is one of the key defining features of neuronal cell types, but is probably also the most difficult type of information to acquire. Revealing the full neuronal morphology is more than a simple labelling problem, because it requires altering the chemical/genetic contents of the neuron to provide sufficient amount of tracers inside the cell. Compared with the classical biocytin cell-filling, ACE employs the intracellular protein synthesis machinery for tracer production, which is advantageous because only a small amount of genetic material needs to be delivered inside the neurons to reveal the full morphology, and damage to the target neurons is better controlled[20,21,23,26–29,32–34]. By electroporating DNA plasmid, ACE demonstrates the capability of perturbing genetic contents of single neurons. Labelled neurons show normal healthiness (Supplementary Fig. 10) and can be subjected to downstream detailed in vitro analyses[38–40]. Viral-vector-mediated gene transfer is a widely used alternative to electroporation, which has been shown to successfully label neurons in vivo[10,35–41]. Due to its stochastic nature of labelling, in most cases viral transfection randomly labels a cluster of neighbouring neurons, making it better suitable for the type of 'labelling a cell' studies. For studies that need to correlate the morphology with functional properties of transfected neurons, approaches of 'labelling the cell' are required. ACE can provide a solution to these 'labelling the cell' scenarios with single-neuron resolution. Besides, because viral injections can often result in labelling of multiple neurons, morphological reconstruction could be difficult. ACE can offer a better alternative in these aspects.

Human experimenters employ a variety of operational tactics in the processes of neuron detection, verification and electroporation[18–21,23–30]. This optimizes yield but causes issues of variability as reflected in literature[21,26,28]. As the first attempt towards solving these problems, we aimed at automating the labour-intensive experimental procedures by standardizing selected sets of optimized procedures and parameters. The effectiveness of our supervised strategy was proven by the demonstrated high success rate ($47 \pm 3$%). It should be noted that during the robot design and implementation we decided to retain the option of manual overriding certain procedures such as the automatic neuron detection and current injection. This is because we determined that a human-in-loop design would be more flexible to allow human intervention when needed for sophisticated in vivo experiments, especially in the hands of experienced experimenters. By doing so, ACE minimizes the variability of experiments and achieves high and stable productivity in labelling of individual neurons in vivo and at the same time, could have versatile applications for different purposes. We recognize the potential for a fully automated system. Based on the success of the current robot, future efforts will be made in order to recapitulate the full sets of successful human operations, and expand them possibly through machine-learning processes, to develop a more autonomous system.

We reported an important use case of ACE, which is to obtain the full morphology of cortical projection neurons recorded

*in vivo*. Future work is needed to extend ACE's application to other brain regions (for example, thalamus), animal species (for example, non-human primates) and behavioural states (for example, awake). We also hope to couple ACE with methods such as trans-synaptic tracing[42,43], or other down-stream methods including slice recording[41] and transcriptomic analysis[39,40,44], so it will be possible to do gene editing, study the connectivity, or correlate full morphology and electrophysiology with transcriptomic properties of single-projection neurons in deep brain structures. The modular design of ACE offers flexibility for various experimental configurations. With proper genetic/chemical materials, ACE can also be used to perform genetic/chemical cellular content manipulations of single-brain cells *in vivo*. The manipulation alone can also provide mechanistic insights on biological processes. In summary, ACE provides a solution to efficiently collect electrophysiological properties and manipulate chemical and/or genetic properties of single-brain cells, even in deep brain structures, and may have broad and scalable applications in various research model systems including non-human primates.

## Methods

**Animals.** Experimental procedures were in accordance with NIH guidelines and approved by the Institutional Animal Care and Use Committee (IACUC) of the Allen Institute for Brain Science. Forty-seven adult C57BL/6J wild-type mice (2–5 months old, both genders) were used in this study.

**Robot design.** The principle of ACE design is illustrated in Fig. 1a. Our implementation of ACE includes following major hardware components:

(1) electrophysiological data acquisition equipment, which is a headstage (CV-7B, Molecular Devices) connected with a patch-clamp amplifier (MultiClamp 700B, Molecular Devices, USA) and digitizer (1440A, Molecular Devices, USA);

(2) A current source (Axoporator 800A, Molecular Devices, USA);

(3) A linear step-motor (PZC200, Newport, USA);

(4) A control box (originally custom-made in house later replaced by Neuromatic Devices, USA);

(5) An electrode holder (1-HL-U, Molecular Device) coupled with a custom-made electrode holder adapter (mill-machined from polycarbonate or 3D printed of polylactide plastics) to accommodate the switch relay (a double pole, double throw relay, Digikey, USA);

and (6) wires and cables connecting individual components.

The headstage, electrode holder with adapter and linear step motor are assembled together and mounted onto a MP-285 4-axis manipulator (Sutter, USA) for initial pipette positioning. All these hardware components are controlled by in-house developed, LabView-based software through a DAQ board (National Instruments, USA) installed on a PC (Dell, USA). The control software features a graphical user interface and integrates parameter setting, experiment logging and operation visualization including pipette advancement, neuron detection/verification and electroporation (Supplementary Fig. 1, also see Supplementary Movie 1).

**Software control and automation.** Glass micropipettes filled with desired pipette solution were manually placed over the surface of the cortical regions of interest. ACE initially measured the pipette resistance ($R_p$) by averaging the current responses to 30 test pulses of $-5$ mV at a high positive pressure ($\sim 150$ mbar, can be adjusted by the user) applied to the pipette. $R_p$ values too high ($>20$ MΩ, indicating pipettes were clogged) or too low ($<3$ MΩ, indicating broken pipettes) were automatically rejected and the trial (penetration) was terminated. With normal $R_p$ values ACE drove the pipette linearly into the brain to reach the starting position set by the user ($\sim 500$ μm underneath the pial in our experiments). The pressure was gradually reduced to the low positive pressure ($\sim 15$ mbar) and $R_p$ was re-measured and compared with the value measured at brain surface to judge whether the pipette was clogged or not. Dramatic increase or decrease of $R_p$ usually indicated a clogged or broken pipette, respectively, both of which would trigger trial termination and the pipette would be automatically retracted to the brain surface and replaced.

If the $R_p$ change was within the tolerable range, ACE would search for single neurons in the brain by driving step advancements (2 μm step per second) of micropipettes with the low positive pressure ($\sim 15$ mbar) maintained to the pipette tip (Fig. 1b,c). At each step $R_p$ was calculated by averaging the current responses to 10–5 mV test pulses. To work in deep brain structure, ACE detected neurons purely electrically (thus blindly) by monitoring the change of $R_p$. Consecutive and monotonic supra-threshold increase of $R_p$ (a relative threshold of 8–10% was used in current study, can be modified by the user) would trigger a detection event. A typical detection occurred within 4–6 μm (Fig. 2a2). Upon detection the low

positive pressure was released, and $R_p$ was compared before and after the pressure release for an initial quality-check of the detection (Initial Detection QC). Due to the elasticity of cytoplasma membrane, further increase of $R_p$ after pressure release suggested 'good' detections. Resistance measured at the atmospheric pressure represented the seal formation between the plasma membrane and the pipette tip. It consisted of the original $R_p$ and the resistance from the interaction between the membrane and glass wall of pipette tip ($R_{cleft}$). Dramatic drop of $R_p$ invalidated the detection, and in these cases, high positive pressure was applied for 5 s to 'clean' the tip and ACE continued to search neurons under low positive pressure, unless the lower limit (1,500 μm underneath the pial in our experiments) was reached. No negative pressure was ever applied to the pipette in order to facilitate the detachment of micropipette after recording or electroporation. Spontaneous activity was recorded for 1–2 min of recording (equivalent to a mean firing rate of $>0.008$ Hz). Appearance of spontaneous action potentials (spikes) verified the presence of a neuron. In case that no spontaneous spikes occurred during the 1–2 min of recording, the detection would be verified by injecting various amount of DC current (10–200 nA, 2 or 5 ms duration, either polarity, single pulse) to evoke spikes. See the Current Injection section for details. Increase of firing immediately after current injection confirmed a neuron detected. Failure of evoking spikes after current injections mainly suggested false positives, which could be glial cells, blood vessels or clogged pipettes. The penetration was then considered as a 'non-neuronal detection' and pipette would be retracted to the surface of the brain. Therefore, our algorithm greatly improved the detection performance by identifying false positives and misses (inactive neurons, Fig. 2c). After the verification, spontaneous activity and/or evoked responses could be recorded, depending on the experimental design. Then using the relay-based switching mechanism, the experimenter could switch between electrophysiology and SCE to manipulate (label) the cell recorded. For electroporation, a train of unipolar (1–2 ms pulse width at 50 Hz for 0.3–2 s with voltage ranging from $-1$ V to $-40$ V) pulses was delivered through the pipette tip. ACE also provided the option to electroporate by current injections through the headstage (upto 200 nA). After electroporation, ACE retracted the micropipette back to the brain surface for next trial. The same pipette could be reused after being retracted to the brain surface for multiple penetrations if dye efflux could be observed, indicating it's not clogged. The entire process was controlled by custom-written LabView software, which is publicly available as the Supplementary Software of this paper. It should be noted that the control software retains the option of manual override of automatic neuron detection, because we think a human-in-loop design could be useful for experienced experimenters to give human intervention in case needed.

**Surgery.** Mice were anesthetized with Isoflurane (1–1.5% in $O_2$) and placed onto a stereotaxic device (Kopf). Body temperature was monitored and maintained at 37 °C with a feedback controlled animal heating pad (Harvard Apparatus, USA). With aseptic surgical procedures, a circular craniotomy ($\sim 4$ mm in diameter) was performed over the primary visual cortex (V1) centring on 1.25 mm anterior and 2.25 mm lateral to the Lambda. In most of the mice another craniotomy was conducted over the primary motor cortex (M1) or anterior lateral motor area (ALM) based on their coordinates, respectively. Thus we could compare cross-region performance of ACE within and between animals. Small pilot holes were made in the dura to facilitate the penetration of the glass micropipettes. Then the craniotomy was covered with sterile ACSF. In case that durotomy was performed, a thin layer of low-melting point agarose (1%, Sigma-Aldrich) was applied over the craniotomy. The mouse was transferred to the experiment set-up for electrophysiology/electroporation. During the entire course of imaging/recording, animals were kept warm and appropriately anesthetized.

**Electrophysiology.** Long-shank borosilicate microelectrodes (glass micropipettes) were obtained by adjusting pulling parameters to reduce possible damage to the brain tissue. The pipettes were filled with either K-gluconate based internal solution containing (in mM): K-gluconate 125, NaCl 10, HEPES 20, Mg-ATP 3, Na-GTP 0.4 in ddH$_2$O; 290 mOsm; pH 7.3, or saline (0.9% NaCl in ddH$_2$O), or ACSF (in mM: NaCl 126, KCl 2.5, NaH$_2$PO$_4$ 1.25, MgCl$_2$ 1, NaHCO$_3$ 26, glucose 10, CaCl$_2$ 2.4, in ddH$_2$O; 290 mOsm; pH 7.3). One of the following plasmid was included in the pipette solution: CAG-EGFP, CAG-TdTomato or pAAV-EF-1α-EGFP (0.05–0.3 μg μl$^{-1}$). The pipette solution was filtered with a 0.2 μm syringe filter before use and the performance was similar. Some of the plasmid was purchased from GenScript. A red fluorescent dye (Alexa-594, 250 μg ml$^{-1}$) was always included to visualize the pipette tip and indicate the tip openness. The pipette was then installed onto the MultiClamp 700B headstage that was mounted onto a MP-285 4-axis manipulator (Sutter, USA) and positioned over the brain surface. The tip of the glass pipette was placed right above the dura matter within ACSF using a surgical microscope. Pipette quality was initially checked visually by ejected dye. In all experiments the pipette was advanced vertically into the brain through the craniotomy over the cortical regions of interest to search for neurons (Fig. 1a). Penetrations were done in a grid with sufficient spatial separation ($\sim 200$ μm) between penetrations to facilitate the identification of labelled neurons. For electrophysiological recordings, signals were amplified with a Multiclamp 700B, digitized with a Digidata 1440B, acquired using the pClamp ver10 software (Molecular Devices) and stored on a PC (Dell, USA). In this study, for simplification only cell-attached recordings were conducted using 3–12 MΩ pipettes.

Spontaneous activity was recorded for at least 1–2 min. For visually evoked responses, whole-screen drifting gratings were presented with a calibrated LCD monitor on the contralateral side showing 8 orientations (45° increment), 3 spatial frequency (0.02, 0.04 and 0.08 cycle per degree) and 1 temporal frequency (2 Hz).

**Current injection.** Current injections were done in detection events that passed the initial detection QC but showed no spontaneous spikes (Supplementary Fig. 3). To determine the minimal amount of current needed to evoke spikes, we first conducted current injections manually in 61 penetrations. In brief, current pulses (square wave, 2 ms width, either polarity) were delivered through the headstage to the pipette. The amplitude of current pulses was gradually increased from 10, 20, 50, 70 (or 80), 100, 120, 150 (or 140) upto 200 nA, with $\geq 1$ s inter-pulse interval. The signal within 1 s post current injection was thresholded at 3–5 s.d. to detect evoked spikes. This is to approximate the minimal amount of current that could reliably fire an inactive neuron without unnecessary damage to its cellular membrane. Current injection was terminated when spikes were evoked or current reached 200 nA, which came first. On average $106 \pm 6$ nA current (mean ± s.e.m., $n = 61$, median: 100 nA, range: 10–200 nA, Supplementary Fig. 3) was needed to evoke spikes. And we found evoked spiking was reversible, indicating limited, if any, damage to the cell (Figs 2 and 4, also see Supplementary Fig. 7). Although attempts were made to automate the current injection for neuron verification by mimicking the manual current injection procedure, we found it was difficult to predict the minimal amount of current required evoking spikes for individual detections by either $R_p$, or the seal resistance ($R_p + R_{cleft}$), or the percentage of $R_p$ increase. The real current measured showed no or weak correlation with $R_p$ or $R_{cleft}$, respectively, and had a broad range of variation between neurons (Supplementary Fig. 4). These data exemplified that the membrane dynamics in response to current injections is currently less known so simple automation that mimics manual current injection may offer little advantage over manual trials. Thus for current injections we kept the human-in-loop option. In our experiments we chose to conduct current injections manually.

**Electroporation.** For in vivo SCE, an Axoporator 800A (Molecular Devices USA) was used to deliver 500 μs pulses at 50 Hz for 0.3–2 s duration at various voltage amplitude (see text for details). About 9–15 penetrations were conducted per animal. At the termination of the experiment the craniotomy was covered with silicone (Kwik-cast) and mice were recovered from anesthesia and allowed to survive for 4–7 days before transcardial perfusion for histology. For in vivo whole-cell recording and biocytin filling ($n = 1$ mouse), biocytin (0.3% w/v, Sigma-Aldrich) was included in the internal solution. Blind whole-cell recording was manually conducted as described previously. At the end of the filling, the pipette was slowly retracted to reseal the membrane of the cell patched.

**Histology and imaging.** The animal was perfused with 4% paraformaldehyde (PFA) in chilled saline through the heart and the brain was removed from the skull. The brain was post-fixed overnight in 4% PFA then transferred to 10% sucrose until it sank. Coronal sections (100 μm) were prepared using a microtome. Sections were mounted using VectaShield, and imaged using a Leica SP8 confocal microscope. Native fluorescence image Z-stacks were collected with a 10 ×, 0.4 NA dry, 20 × 0.85 NA oil-, 40 ×, 1.3 NA oil- or 63 ×, 1.4 NA oil-immersion objective at 2 μm step size. For revealing morphological detail, Z-stacks were tiled to cover the dendritic arbour across adjacent slices. These stacks were stitched using an ImageJ stitching plug-in and aligned with the Vaa3d software (www.vaa3d.org). Images of the larger field of view were taken with a 10 ×, 0.4 NA dry objective. For biocytin filled cells, the slices were processed with Biocytin-Streptavidin reaction then imaged under the Leica SP8 confocal microscope.

**Data analysis.** For each experiment we extracted the time, depth, $R_p$ and type of the operations from the corresponding ACE log. To measure ACE performance (Fig. 2 and Supplementary Fig. 2), for each experiment we had:

Total trials (penetrations), which include all the trials conducted to a cortical region of interest; detection trials, in which detection events were successfully triggered; trials with neuron verified: detection trials with neurons confirmed by either spontaneous or current-injection-evoked spikes. So, the number of total trials = number of detection trials + number of trials in which the pipettes were clogged or broken; and the number of detection trials = number of trials with neuron verified + number of trials with non-neuronal detections. In addition, we defined: detection prevalence = number of detection trials/number of all trials; detection prevalence = number of detection trials/number of all trials; detection precision = number of trials with neuron verified/number of detection trials; false-positive rate = number of non-neuronal detection trials/number of detection trials.

For each trial (penetration), we also calculated the detection time ($t_{detection}$) and total trial time ($t_{total}$). $t_{detection}$ was defined as the time lapse from the start time of the trial to the time when the final detection was successfully triggered. Thus for each trial:

$$t_{detection} = t_{final\ detection} - t_{start},$$

$t_{total} = t_{end} - t_{start}$, where $t_{end}$ is the end time of the trial.

Moreover, we compared these values across brain regions and animals.

For electroporation only experiments (as shown in Fig. 3), because no neuron verification by current injections was done the success rate was calculated for each experiment as the ratio between the number of penetrations with neuron recovered and total number of penetration. For electroporation + Ephys experiments (Fig. 4), the success rate was calculated for each experiment as the ratio between the number of penetrations with neuron recovered and the number of penetration with neuron verified. Brains showing signs of damage or infection were excluded from the analysis.

Labelling single neurons investigated in vivo, which is our ultimate goal, is an extremely complicated experiment, here we took a typical 'divide and conquer' strategy in experiment automation and parameter optimization. We recognized that many factors might contribute to the final results, but in general we considered it reasonable to decompose the entire experiment into four major processes:

(1) neuron search;
(2) electrophysiology;
(3) electroporation and
(4) neuron survival after electroporation.

In theory, (5) some hidden factors should also be included. Each process can then be subdivided into critical sub-processes. In this effort, if we denote the success rate of each process as $p$, we have:

$$p(\text{experiment}) = p(\text{neuron search}) * p(\text{electrophysiology}) * p(\text{electroporation})$$
$$* p(\text{neuron survival}) * p(\text{hidden factors}), \text{ and } p(\text{experiment}) \text{ should empirically}$$
$$\text{equal to (No.of neurons recovered)/(total penetration)};$$

And because we have identified key sub-processes in neuron search, we have:

$$p(\text{neuron search}) = p(\text{detection event}) \times p(\text{neuron verified in detection events})$$
$$= 0.9 \times 0.9 = \sim 0.8;$$

In the current study all efforts were made to maximize $p(\text{experiment})$. $p(\text{electrophysiology})$ is close to 1 and we believe we have nearly optimized $p(\text{neuron search})$. Unfortunately $p(\text{electroporation})$ can't be easily determined in vivo, but we could estimate its upper limit using the data from in vivo two-photon targeted SCE experiments, which is known to be ~90%. Considerable amount of efforts had been made to approach the actual $p(\text{electroporation})$ to this upper limit by a combination of improvements including optimizing pipette geometry, electroporation voltage, plasmid concentration, pipette solution type and so on as stated above. Similar efforts were also made to increase the chance of neuron survival post electroporation—$p(\text{neuron survival})$ to improve the final success rate $p(\text{experiment})$.

Healthiness of neurons was assessed during four major experiment processes including: neuron search; electrophysiology; electroporation and neuron survival after electroporation. Long-shank pipettes were used to reduce possible damage to the brain tissue. Neuronal healthiness during neuron search was carried out by our pipette and detection QC (see the Software Control and Automation section). We also assessed physiological healthiness by examining the spontaneous and visually evoked firing activity, such as the action potential (spike) shape, reversibility of firing in response to current injection, orientation-tuning curves and so on (Fig. 2b). Due to the depth of these neurons, it was not possible to find labelled neurons again and record their electrophysiological responses in vivo. Healthiness of neurons post electroporation/infusion of DNA plasmid has been investigated by multiple research groups in tadpole tectum[23,32], mouse barrel cortex[34] and visual cortex[20,21,29], both functionally and anatomically at various timescales. These studies, especially those two-photon guided work, provide evidence that negative impact of the electroporation process on neuronal healthiness would be limited if any. As for those successfully labelled neurons, in the present study their healthiness was mainly assessed by post-mortem histology. In brief, brain tissue containing labelled neurons was carefully inspected for possible tissue damage, as we have shown in Figs 3 and 4, Supplementary Figs 5–9, and DAPI staining was conducted to examine possible neuronal necrosis (Supplementary Fig. 10). No significant tissue health issues were noticed with either method.

Reconstruction of the full morphology of recovered neurons will be included in separate studies. T-test was used to detected statistical significance.

**Data availability.** The data that support the findings of this study are available from the corresponding author upon reasonable request.

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

## Acknowledgements

We thank Adrian Cheng, Elliot Mount, Ulf Knoblich and Suvro Datta for technical support. We thank Bosiljka Tasic, Thuc Nguyen, Maya Mills, Hong Gu and Linda Madisen for providing and preparing the initial CAG-EGFP and CAG-tdTomato plasmid. This work is supported by the Allen Institute for Brain Science, and an NIH grant (R01EY023173) to H.Z. and C.R.F. We thank Hannah Krakauer for editing the manuscript. The authors wish to thank the Allen Institute founders, Paul G. Allen and Jody Allen, for their vision, encouragement and support.

## Author contributions

L.L. initiated the study. L.L. and B.O. performed the experiments. L.L. and W.S. developed the software. B.O., for example, and T.L.D. conducted histology, imaging and plasmid preparation. W.S. and C.R.F. modified the Autopatcher box used in the experiments. L.L. analysed the data. L.L., C.K. and H.Z. supervised the project. L.L drafted the manuscript with inputs from all authors.

## Additional information

**Competing interests:** The authors declare no competing financial interests.

