## [Peer Review File · Nature Communications]

Reviewers' comments:

Reviewer #1 (Remarks to the Author):

The authors present a strategy for automating in vivo single-cell electroporation and electrophysiology experiments. They develop metrics and algorithms for maximizing the success rate for semi-automated 'blind' loose-patch and whole-cell recordings, and for juxtacellular electroporation experiments, demonstrating the use and efficacy of their approach with recordings and transfections of deep-layer pyramidal neurons in anaesthetized mice.

This manuscript is part of a series of papers describing different automation techniques for in vivo physiology (e.g. Long et al Sci. Reports 2015, Wu et al J Neurophys 2016). It is therefore part of a larger effort for scaling up laboratory techniques to enable high-throughput physiology experiments, even in labs which have minimal experience in this field. The advance represented in this paper is useful and important, though it is not sufficient to allow full automation of such experiments (see point 8 below). The success rates the authors present appear to be sufficiently high to make this a technique that will attract the interest of a large numbers of labs interested in using these techniques. However, more clarity is required both for the description of the technique and the definition of success rates.

Detailed concerns:

1. The authors do not sufficiently describe their methodology in several respects. From the main text, it is very difficult to understand what the final electrophysiological recording configuration is, and also how they get there. The figures suggest that they achieve a loose-patch or cell-attached recording configuration and not a whole-cell recording, but this needs to be stated explicitly, given that spiking alone contains less information than access to subthreshold activity. The use of current injections for improvement of automated neuron detection also is not clear. The authors state that using current injections to test whether a putative neuron can spike improves their detection rate, but the exact logic is not explained: When and how are these current injections applied? What is the algorithm / the decision tree of the procedure? In the classic autopatcher approach, positive pressure on the pipette is released to establish a gigaseal if changes in the test pulse response indicate that a neuron might be close. Is negative pressure applied here, and if so, in what order with current test pulses?

2. The success rate ultimately determines the usefulness and power of this approach. However there are different conditions, and a few different kinds of success being evaluated in the paper - and the terminology that is defined in contradictory ways in different sections. This leads to confusion. It would help to organize them into a table, or at least clarify the wording. Some relevant questions in this category - which should all be relatively easily addressable with textual changes - include the following:

a) The authors define "hit rate" first as "successful detection of neurons" in the very first sentence of the results. At the end of this paragraph they re-define it as "chance of neuron when detection was triggered." These two things are not the same. The first would have to account for false negatives - when there were neurons that could be recorded or electroporated but they were not detected. Clearly this information is unavailable. The second is the proportion of detection events that represent true hits. The second is a reasonable definition, but it would be best to use only one definition.

b) The authors go on to use this definition of hit rate, as a kind of success rate. However for an ultimate success rate to be calculated, it should be multiplied by the rate at which detection events were triggered, which they report is 90%. This won't change the results but it is the more relevant number for the reader as an ultimate success rate quantification. So the overall success rate in getting recordings from neurons would be 80% - is that correct?

c) The authors call the electroporation success rate in figure 3G simply 'success rate'. It would help to call this "electroporation success rate" and not the total success rate, which would have to include the rates of the previous steps and the following steps.

d) What is the rate at which detection events were triggered (# of penetrations with detection / # all penetrations) when using K-gluconate? It is unclear if in this condition the detection probabilities changed.

3. Given that the ultimate utility of the proposed method will lie in the combination of extensive functional characterization and morphological reconstruction, it is crucial to systematically test whether long recordings prior to electroporation affect the overall success rate. Does the duration of the recording affect the success rate of subsequent electroporation?

4. The authors state that electroporated neurons showed 'normal healthiness', but do not provide any measure for this. They should provide electrophysiological controls to demonstrate this. They should also provide some analysis of the sensory responses they obtained to demonstrate the health of their recordings.

5. The authors state that their electroporation success is 3 times higher than most manual blind studies. They cite three papers. One is done on non-human primates, which is not a reasonable basis for comparison. Oyama et al achieve 40% electroporation success in their best condition, which seems comparable. The last citation shows substantially lower success rates. So the conclusion "about 3-fold higher than most blind manual studies" is overstating the case.

6. Regarding viral vector mediated gene transfer, which is an alternative strategy for getting single neurons or sparse populations to express transgenes, the authors state: "viral transfection is a stochastic process, making the morphology unlikely to correlate with functional properties of transfected neurons." It is not clear what this means and how blind electroporation solves the problem. They also state: "toxicity poses another issue." Current

AAV based techniques are fairly widely used, and while toxicity becomes an issue for studies involving long durations of expression, there certainly is a sufficiently large enough time window within which to get full morphologies and healthy cells. The final problem cited for the viral technique is that it is hard to get single neurons. This is certainly true. But on the other hand injecting virus is substantially easier than single cell electroporation, and several groups have worked out reliable techniques for getting sparse labeling with viruses. This is not to say that electroporation isn't needed, but the authors paint an unnecessarily bleak picture of this strategy.

7. The Discussion should normally be where the authors place their method in the context of the existing literature. However, there are only two citations in the entire Discussion section. To give an example, the authors tout the benefits of their approach for single-cell transfection: "By electroporating DNA plasmid, ACE demonstrated the capability of perturbing genetic contents of single neurons". But this is not a novel feature of their method - in vivo single-cell electroporation (blind or visually targeted) has been performed by many previous studies, none of which are cited here. The authors should do a more scholarly job of referencing the prior literature and explaining the advantages (and disadvantages) of their approach in the Discussion section.

8. The title "High-yield single-cell characterization and perturbation of brain neurons in vivo" is very vague - it gives no indication of the method being used, what kind of "characterization" and "perturbation" is being done, etc.

9. The long-term aim of the authors is to reach a stage of automation that requires minimal supervision and minimal training of users. However, the paper does not evaluate - or even properly acknowledge - key technical issues that prevent complete automation of these experiments. An important example is surgeries: a critical factor in the success of electroporation experiments is the quality and cleanliness of the surgery. It takes humans time to get good enough at these surgeries. Second, operation of the automated device still requires that the materials involved be generated and handled by trained personnel. Learning how to make good internal solution (which is required for the success rates reported here) is a time-honored rite of passage in physiology labs around the world, meaning it takes training. Making patch pipettes of the correct size and resistance also requires familiarity. The list goes on, and the ultimate picture is that trained professionals who specialize in these techniques are still necessary for use of the automated machine. Overall this lessens the impact of the advance described here, since staff with the necessary training to master all of the other required elements (some of them listed above) will probably not need the use of automation of this particular step. The authors should consider and comment on this wider issue.

Minor comment:

In Fig. 3G and 4C, comparing the "success rates" of the automated and manual approaches, it would be helpful to label these directly on the figure (rather than obliging the reader to visit the figure legend to figure out what is going on).

Reviewer #2 (Remarks to the Author):

The authors describe an automated system for combining single-cell electroporation and cell-attached patch recording in an in-vivo preparation. They validate their system on deep-layer cortical neurons in anesthetized mice, achieving success rates comparable to those seen in manual experiments. The paper should be useful to many researchers interested in using single-cell measurement, perturbation, and reconstruction in vivo.

I have some comments/suggestions.

Neuron identification/sealing: (A) The system identifies neurons and obtains seals in a manner very like that described in earlier work by some of these authors (Kodandaramaiah et al., Nat. Methods 2012). Yet there are subtle differences that aren't noted or explained. (1) Kodandaramaiah et al. use 800-1000 mbar when pipettes enter the brain. Here the corresponding pressure is only 200 mbar. Why? Does the higher pressure damage tissue or eject too much intracellular solution into the extracellular space? (2) Kodandaramaiah et al. describe a multi-step process of detecting a neuron (with an absolute criterion that depends on the initial pipette resistance). Here the procedure seems to be simply an 8-10% increase. Again, why? (B) What does the system do if the detection verification test fails (no spontaneous or evoked spikes)? Does it withdraw the pipette or does it keep moving to find another neuron? (C) Can the pipettes be reused since they're not forming tight seals? That is, can one record/electroporate a neuron, detach, move laterally, and search for another neuron?

$R_p + R_{cleft}$: How much does the value of the total resistance ($R_p + R_{cleft}$) affect the quality of the cell-attached recording? The manuscript is rather vague about how spikes are detected. From the figures (for example, Fig. 2), the identification does not appear to be ambiguous, but the authors should describe precisely what it was (absolute threshold, threshold in standard deviations, maximum-minimum spike width, canonical spike shape, etc.). They should also test whether spike detection depended on $R_p + R_{cleft}$. Without that test, it's hard to know what Suppl. Fig. 2 is showing us.

SCE: A single procedure (12 V, 1-2 ms pulses at 50Hz for 0.3-2.0 s) was used for electroporation. Was this procedure ever varied to determine whether it was indeed optimal? And did the success rate of SCE depend on the quality of the seal ($R_p + R_{cleft}$)? (This second question is the one that I think is most important and that many researchers would like to know.)

Multiple penetrations: The value of automating these processes, as the authors note, is the potential to maintain high throughput. One thing that limits high throughput is tissue damage because of multiple penetrations. Do the authors have any data on this question? Their visual cortical craniotomies are quite large (4 mm diameter), suggesting that by spacing electrode penetrations the authors could use many, many electrodes per craniotomy. Is this right? Put another way, what limits the number of neurons and neuronal

locations that can be obtained with this system?

Automation: The authors should describe more clearly what parts of their system are automated and what parts are not. They note that the current injection / neuron identification part includes some leeway for the experimenter to intervene, but it's not clear what they mean. (By the way: "man-in-loop" should be "human-in-the-loop". The second term is what appears in the engineering literature nowadays, and so is the preferred term of art.) From the WMV file (1932465), it appears that interaction with the Multiclamp Commander is not automated. Why not? It should be since the Commander is a piece of software.

Equipment: How much flexibility is there in the choice of equipment? For example, say that a researcher had a manipulator from Luigs-Neumann? How difficult would it be for them to adapt the authors' system and how quite would they do it? How about people who have DAQ devices from National Instruments or HEKA? Is Digidata required for some reason? And what about the Neuromatic control box? The authors write that in their earlier paper the functions of the control box were custom made but that they are now available in commercial form through Neuromatic Devices. That's fine, but what exactly are those functions? Those readers who have read earlier work from the Forest group will know, but this publication – which, like all publications, should be as standalone as possible – should spell it out. A related issue: how expensive is the Neuromatic control box? For a paper of this sort to have value, it must persuade other researchers to try out the methods. A big stumbling block is money. As gauche as it may be to point this out: if the box costs \$1000, many people will be tempted to give it a go; if it costs \$10,000, many of those same people must stop and think hard; and if it costs \$100,000, most of the remainder will immediately stop reading the paper. The Neuromatic Devices website provides no guidance on this point – only a link to communicate with (presumably) sales staff. The alternative is a custom-made device. How hard and expensive would that be?

I also had some minor comments on figures / supplementary material.

1) In Fig. 1A, the Control Box also provides pressure to the pressure port of the electrode holder. This should be indicated in some manner.

2) In Fig. 2A2, there is an initial drop in R_p from 10.2 to 9.4 MOhms. Is that simply a consequence of changing the positive pressure from 200 mbar to 20 mbar? If so, that should be noted in the figure caption.

3) The AVI file (1932466) in the supplementary material won't play. I tried it on multiple computers but always got an error.

Reviewer #3 (Remarks to the Author):

The manuscript by Li and colleagues describes a robotic platform that aims to enhance the efficiency of performing automated single cell electrophysiology and electroporation experiments in anesthetized mice. The authors suggest that these technologies will facilitate investigation of neuronal structure-function relationships and cell-type classification. In general the data is well presented and logical providing a series of optimization steps to the already well-established process of automated in vivo patching. What is not clear are the novel aspects of their approach given that a number of different groups have already established the majority of the methods described? As described, the data appear as incremental steps in the optimization of existing methodologies rather than a completely novel approach.

The first rather misleading aspect of the manuscript is the title. As read, it suggests that the authors use novel technologies to perturb some aspect of neuronal activity in order to better understand function. However, the authors do not show data where cell function has been manipulated, instead they express EGFP, which is not a perturbation, rather a knock-in with hopefully no physiological effect on function.

The authors then suggest that the technical advance provided by their robotic Automated single-Cell Experimenter (ACE) and Single-Cell Electroporation (SCE) is a significantly higher yield than previously published methods. But the authors compare their statistics with a publication from the Boyden lab that focused on attaining whole-cell patch-clamp recordings, where it is likely that 'failed' patches were recorded in the cell-attached configuration (Kodandaramaiah et al., 2012). Also, the approach used negative suction to attain $>1\text{G}\Omega$ seals which is very different to the 'atmospheric pressure' release approach used in the present study. This is a much simpler method to attain cell-attached recordings, which presumably can be achieved using any of the existing 'autopatch' methods. The authors then compare the efficiency of their method with a paper that focused on patching in awake, behaving mice (Desai et al., 2015). The brain displacements that occur in awake mice significantly reduce the possibility of attaining cell-attached or patch recording configurations, so it is rather odd to compare recording efficiency in two completely different preparations. Therefore, I don't really see this as a significant technical advance, rather an optimization of existing technologies.

The additional 'novel' features of the robotic system are the 1) automated AP generation during cell-attached configuration to enhance cell detection and 2) automated electroporation of exogenous substances. Unfortunately, both of the approaches can be simply automated via existing software packages when performing manual 'blind-patch' recordings in vivo. So again it is not clear what advantage is gained from implementing these technologies. Moreover, the ability to deliver plasmids and express EGFP via in vivo electroporation has been shown before (Kitamura et al., 2009; Rancz et al., 2011) so again the data appears to be a subtle optimization of existing technologies.

The authors state that intracellular infusion of biocytin during patch recordings – seen as the gold standard technique for morphological reconstructions in vivo – may not capture the full morphological extent of a neuron. What evidence is there to support this claim? To what extent is EGFP expression more efficient – this should be shown as experimental data. The

authors then seem to contradict themselves by stating "It has been shown that cell-labeling with fluorescent proteins is comparable to biocytin-cell filling". I agree with the authors that single cell electroporation spatially restricts protein expression but the number of 'ghost cells' when using biocytin can be significantly reduced or controlled by reducing the total number of penetrations per preparation. In the paper the authors perform 9-15 penetrations to infragranular layers – given the standard geometry of patch pipettes, this high number of penetrations will inevitably damage the cortex so it is not surprising to find that many non-recorded 'ghost' cells display biocytin staining. Simply reducing the total number of penetrations will significantly reduce the number of 'ghost cells'.

Itemized Response to Reviewer 1:

Major issue # 1) The authors do not sufficiently describe their methodology in several respects. From the main text, it is very difficult to understand what the final electrophysiological recording configuration is, and also how they get there. The figures suggest that they achieve a loose-patch or cell-attached recording configuration and not a whole-cell recording, but this needs to be stated explicitly, given that spiking alone contains less information than access to subthreshold activity. The use of current injections for improvement of automated neuron detection also is not clear. The authors state that using current injections to test whether a putative neuron can spike improves their detection rate, but the exact logic is not explained: When and how are these current injections applied? What is the algorithm / the decision tree of the procedure? In the classic autopatcher approach, positive pressure on the pipette is released to establish a gigaseal if changes in the test pulse response indicate that a neuron might be close. Is negative pressure applied here, and if so, in what order with current test pulses?

Response: We completely agree with the reviewer that providing sufficient amount of technical detail will facilitate the understanding and may lead to future adoption of our invention by other laboratories. We apologize for any ambiguity or misunderstanding that may have been caused. In our original submission a rather detailed description of the operational procedure of ACE had already been included in the online **Supplementary Information**, but not the main text, primarily due to the page limit requirement for technique papers. The main text contained a brief summary, and the Supplementary Information actually elaborated the processes employed by ACE, such as the neuron detection, initial quality check of the detection, neuron verification etc., as well as the rationale behind these operations, and it seems to us the Supplementary Information could address most of the questions the reviewer raised here. To be more specific, as for whether negative pressure was applied, the **Software Control and Automation** section of the Supplementary Information stated “No negative pressure was ever applied to the pipette in order to facilitate the detachment of micropipette after recording or electroporation” (page# 3, paragraph 1). In regard to when and how current injections were applied, an entire section of **Current Injection** had been developed for this purpose (Page# 4, paragraph 2). We acknowledge that the recording configuration may be confusing (but see our main text page #4, paragraph 1, first sentence), probably because the sentence of “In this study, for simplification only cell-attached recordings were conducted using 3–12 MΩ pipettes” was put in the **Electrophysiology** section of the Supplementary Information (page# 4, paragraph 1). To minimize possible confusion, in the revised manuscript we clarified these points in the main text. We thank the suggestion for visualizing the decision tree, which we think is just brilliant. Accordingly we have added a figure (**new Suppl Fig. 3**) to the revised manuscript.

Major issue # 2) The success rate ultimately determines the usefulness and power of this approach. However there are different conditions, and a few different kinds of success being evaluated in the paper - and the terminology that is defined in contradictory ways in different sections. This leads to confusion. It would help to organize them into a table, or at least clarify the wording. Some relevant questions in this category - which should all be relatively easily addressable with textual changes - include the following:

a) The authors define "hit rate" first as "successful detection of neurons" in the very first sentence of the results. At the end of this paragraph they re-define it as "chance of neuron when detection was triggered." These two things are not the same. The first would have to account for false negatives - when there were neurons that could be recorded or electroporated but they were not detected. Clearly this information is unavailable. The second is the proportion of detection events that represent true hits. The second is a reasonable definition, but it would be best to use only one definition.

Response: We are glad to see the reviewer pointed out that success rate has a context-dependent meaning. This is inevitable because such an *in vivo* single-neuron electroporation experiment consists of multiple key processes, and some processes can be subdivided into several sub-processes (A typical 'divide and conquer' strategy, also see our more elaborated response to comment **2.c**). Thus the success rate of each (sub)process needs to be quantified and optimized, in order to maximize the final experiment success rate. We agree with the reviewer that to measure the 'hit-rate', information of false negatives (or 'miss') should be available. We again agree with the reviewer that the ground truth of true positives or false negatives is unavailable for 'blind' experiments in deep brain structures. In the current study we think we reduced the occurrence of misses because 1) a low detection threshold (8%-10% increase of the original Rp) was used, which will reduce the chance of passing a neuron without detection; and 2) detection events were successfully triggered in all penetrations where the pipette was not clogged. Thus by reducing false negatives we think the 'hit-rate (of neurons)' was improved. Then we empirically measured the precision of detection by using the 'chance of neuron when detection was triggered'. We thank the reviewer pointed out this and we added more clarification to the revised manuscript.

b) The authors go on to use this definition of hit rate, as a kind of success rate. However for an ultimate success rate to be calculated, it should be multiplied by the rate at which detection events were triggered, which they report is 90%. This won't change the results but it is the more relevant number for the reader as an ultimate success rate quantification. So the overall success rate in getting recordings from neurons would be 80% - is that correct?

Response: The short answer is yes but the number will be less meaningful. In the current study, we provided both numbers because they are two related but different metrics: prevalence and precision. For prevalence we quantified the detection probability exactly as the reviewer pointed out – "the rate at which detection events were triggered, which they report is 90%". For the latter we quantified the chance of neuron in detection events and used it as the empirical success rate of neuron detection, because detection events consist of neuronal and non-neuronal detections. The calculation of success rate (of neuron detection) here was approximated by the "chance of neuron when detection was triggered", as we have explained in our response to comment **2.a**. By this definition (please also see the **Data Analysis** section of the **Supplementary Information**), rejected penetrations (10± 2% of total penetrations) were not, and shouldn't be included in the calculation simply because no detection events were ever triggered in these penetrations. If rejected penetrations were included in the base, the number would be 78% (146/188 penetrations, see the main text page# 4 and 5, also Fig. 2C). Penetration rejection is necessary and platform independent, because pipette clogging/breaking/Rp-out-of-range would occur no matter manual and automated approaches were used. In the current study, rejected penetrations consisted of a small portion (~10%) of total penetrations thus we think the success rate we reported here represents a better metric of the performance of our platform.

c) The authors call the electroporation success rate in figure 3G simply 'success rate'. It would help to call this "electroporation success rate" and not the total success rate, which would have to include the rates of the previous steps and the following steps.

Response: We hope we didn't misunderstand this comment since we found it a bit confusing – the number reported in Fig. 3G (~16%) was actually the ratio of total number of neurons eventually recovered (21 neurons) to the total number of penetrations (130 penetrations, page# 6, paragraph 2). Thus we think this calculation gave us the total success rate, as we stated in the manuscript.

We agree with the reviewer that knowing the success rate of electroporation will be informative. Because labeling single neurons investigated *in vivo*, which is our ultimate goal, is an extremely

complicated experiment, here we'd like to take this opportunity to elaborate on our strategy in experiment automation and parameter optimization (also see our more elaborated response to comment **2.a**). It is very true that many factors contribute to the final results we will get, but in general we think it reasonable to decompose the entire experiment into 4 major processes: 1) neuron search; 2) electrophysiology; 3) electroporation and 4) neuron survival after electroporation. In theory 5) some hidden factors should also be included. Each process can then be subdivided into critical sub-processes. In this effort, if we denote the success rate of each process as p , we have:

$$p(\text{experiment}) = p(\text{neuron search}) * p(\text{electrophysiology}) * p(\text{electroporation}) * p(\text{neuron survival}) * p(\text{hidden factors}),$$

and $p(\text{experiment})$ should empirically equal to $(\# \text{ of neurons recovered}) / (\text{total penetration})$;

And because we have identified key sub-processes in neuron search, we have:

$$p(\text{neuron search}) = p(\text{detection event}) \times p(\text{neuron verified in detection events}) = 0.9 \times 0.9 = \sim 0.8;$$

In the current study all efforts were made to maximize $p(\text{experiment})$. $p(\text{electrophysiology})$ is close to 1 and we believe we have nearly optimized $p(\text{neuron search})$. Unfortunately $p(\text{electroporation})$ can't be easily determined in vivo, but we could estimate its upper limit using the data from in vivo two-photon targeted single-cell electroporation experiments, which is known to be $\sim 90\%$. Considerable amount of efforts had been made to approach the actual $p(\text{electroporation})$ to this upper limit by a combination of improvements including optimizing pipette geometry, electroporation voltage, plasmid concentration, pipette solution type and so on as stated in our manuscript. Similar efforts were also made to increase the chance of neuron survival post electroporation - $p(\text{neuron survival})$ in order to improve the final success rate $p(\text{experiment})$.

d) What is the rate at which detection events were triggered ($\#$ of penetrations with detection / $\#$ all penetrations) when using K-gluconate? It is unclear if in this condition the detection probabilities changed.

Response: We just realized only saline and ACSF were included in the text of our original submission when we stated that the neuron detection was independent of types of pipette solutions (page# 5, paragraph 2). We thank the reviewer for pointing this out for us. We didn't notice any significant difference in neuron detection rate when using K-gluconate internal solution, compared with experiments using saline or ACSF. This piece of data with K-gluconate internal solution was now added to the revised manuscript.

Major issue # 3) Given that the ultimate utility of the proposed method will lie in the combination of extensive functional characterization and morphological reconstruction, it is crucial to systematically test whether long recordings prior to electroporation affect the overall success rate. Does the duration of the recording affect the success rate of subsequent electroporation?

Response: We presented our data in Fig. 4 showing that infragranular neurons that we recorded in V1 can be revealed, but we agree with the reviewer on this. In our dataset we had conducted ~ 30 minutes of electrophysiological recording and successfully recovered the neuron recorded (**new Suppl Fig. 8**). We haven't noticed apparent correlation between the length of recording time (≤ 30 minutes) and success rate of labeling. Thirty minutes of recording was chosen because we think it represents reasonably long period of electrophysiological characterization. These results were included in the revised manuscript.

Major issue # 4) The authors state that electroporated neurons showed 'normal healthiness', but do not provide any measure for this. They should provide electrophysiological controls to demonstrate this.

They should also provide some analysis of the sensory responses they obtained to demonstrate the health of their recordings.

Response: Healthiness of neurons is very important to us because we want to reconstruct the full morphology of the cells we recorded. In the present study this was analyzed according to the four major processes mentioned before: 1) neuron search; 2) electrophysiology; 3) electroporation and 4) neuron survival after electroporation, see our response to comment **2.c**. We made long-shank pipettes to reduce possible damage to the brain tissue. Neuronal healthiness during neuron search was carried out by our pipette and detection QC (see the **Software Control and Automation** section in the Supplementary Information). We also assessed physiological healthiness by examining the spontaneous and visually evoked firing activity, such as the action potential (spike) shape, reversibility of firing in response to current injection, orientation tuning curves etc (**Fig. 2B1-3**). Due to the depth of these neurons, it is not possible to find them again and record their electrophysiological responses in vivo. Healthiness of neurons post electroporation/infusion of DNA plasmid has been investigated by multiple research groups in tadpole tectum (e.g. Haas et al., Neuron 2001; Bestman et al. Nat Protoc 2006), mouse barrel cortex (e.g. Pala and Petersen Neuron 2015) and visual cortex (e.g. Kitamura et al., 2009; Judkewitz et al. 2009; Rancz et al. Nat Neurosci 2011), both functionally and anatomically at various timescales. These studies, especially those two-photon guided work, provide evidence that negative impact of the electroporation process on neuronal healthiness would be limited if any. As for those successfully labeled neurons, in the present study their healthiness was mainly assessed histologically. Briefly brain tissue containing labeled neurons was carefully inspected for possible tissue damage, as we have shown in **Figs 3&4, Suppl. Fig. 5-8**, and DAPI staining was conducted to examine possible neuronal necrosis (**new Suppl. Fig. 10**). We didn't notice significant tissue health issues with either method, and we can reconstruct the very distal projections of single V1 L5 neuron labeled with ACE in vivo (see the **Figure for Reviewers**). We have expanded the text and added one more figure (**new Suppl. Fig. 10**) to include this in the revised Supplementary Information.

Major issue # 5) The authors state that their electroporation success is 3 times higher than most manual blind studies. They cite three papers. One is done on non-human primates, which is not a reasonable basis for comparison. Oyama et al achieve 40% electroporation success in their best condition, which seems comparable. The last citation shows substantially lower success rates. So the conclusion "about 3-fold higher than most blind manual studies" is overstating the case.

Response: The primate paper was mentioned to show single-cell electroporation can be used in non-human primates, which could be one of our future directions. We agree with reviewer's comments on the success rate of these published manual electroporation studies. Actually we are quite familiar with these results because in our pilot experiments we compared published strategies we could find in literature to choose and automate the best available in vivo blind single-cell electroporation method and Oyama et al. was certainly one of our top choices. Unfortunately in our hands we couldn't repeat Oyama et al's success even though we rather strictly followed their description (for example using pipettes with comparable resistance and manually advancing or retracting pipettes to have ~30% increase of R_p and ~0.5mV spikes before electroporation). These negative results, which could reflect that we lack the special expertise required for Oyama et al J Neurosci Methods 2013, were not mentioned in our manuscript but we do know other groups had similar experience (for example see Dempsey et al Physiol Rep 2015, page# 8), which we quoted here below: "We adopted an approach similar to that used for electroporation of superficial cortical neurons (Judkewitz et al. 2009; Oyama et al. 2013), in which the pipette is maneuvered such that R_s is increased by 30%, and attempted SCE in 31 neurons in five recovery experiments; no transfected cells were subsequently identified". This indicates actual results could be heavily dependent on experimenters' skills, which is one thing we would like to solve here through standardizing and automating experiment procedures. The median of success rate of blind

electroporation in deep brain regions appears to us in the 10 – 15% range, thus we think our platform represents considerable amount of improvement. With this being said, we have rephrased this sentence in the revised manuscript to avoid overstatement.

Major issue # 6) Regarding viral vector mediated gene transfer, which is an alternative strategy for getting single neurons or sparse populations to express transgenes, the authors state: "viral transfection is a stochastic process, making the morphology unlikely to correlate with functional properties of transfected neurons." It is not clear what this means and how blind electroporation solves the problem. They also state: "toxicity poses another issue." Current AAV based techniques are fairly widely used, and while toxicity becomes an issue for studies involving long durations of expression, there certainly is a sufficiently large enough time window within which to get full morphologies and healthy cells. The final problem cited for the viral technique is that it is hard to get single neurons. This is certainly true. But on the other hand injecting virus is substantially easier than single cell electroporation, and several groups have worked out reliable techniques for getting sparse labeling with viruses. This is not to say that electroporation isn't needed, but the authors paint an unnecessarily bleak picture of this strategy.

Response: There is no question that the viral vector mediated gene transfer is an excellent technique. The key message we would like to convey here can be summarized as "label a neuron vs. label the neuron". Viral injection has been well established to randomly label a set of neighbouring neurons, which does a great job if "label a neuron" is needed. For our research goal here, which is to recover the neuron we record, viral method may not be as suitable as ACE to "label the neuron (recorded)", but this doesn't necessarily mean viral technique is a "bad" technique. We agree with the reviewer's comments on viral strategy, we have rephrased this part to give a more balanced overview.

Major issue # 7) The Discussion should normally be where the authors place their method in the context of the existing literature. However, there are only two citations in the entire Discussion section. To give an example, the authors tout the benefits of their approach for single-cell transfection: "By electroporating DNA plasmid, ACE demonstrated the capability of perturbing genetic contents of single neurons". But this is not a novel feature of their method - *in vivo* single-cell electroporation (blind or visually targeted) has been performed by many previous studies, none of which are cited here. The authors should do a more scholarly job of referencing the prior literature and explaining the advantages (and disadvantages) of their approach in the Discussion section.

Response: We thank the suggestion by the reviewer. In our original submission we thought since we had compared existing methods for labeling recorded neurons in the Introduction to explain our motivation to develop ACE, in the Discussion we didn't repeat the comparison. In this revised manuscript we have updated the Discussion to include more in-depth comparison and cite more published studies. We are fully aware that many studies have been conducted but we can only cite a limited number of references. We apologize if there are important published studies that we should cite but omitted. If the reviewer kindly suggests some of those important studies, we will be happy to include them.

Major issue # 8) The title "High-yield single-cell characterization and perturbation of brain neurons *in vivo*" is very vague - it gives no indication of the method being used, what kind of "characterization" and "perturbation" is being done, etc.

Response: We thank the reviewer for the suggestion. We have changed the title to "A robot for high-yield, *in vivo* single-neuron electrophysiology and full morphology" for better clarity.

Major issue # 9) The long-term aim of the authors is to reach a stage of automation that requires minimal supervision and minimal training of users. However, the paper does not evaluate - or even

properly acknowledge - key technical issues that prevent complete automation of these experiments. An important example is surgeries: a critical factor in the success of electroporation experiments is the quality and cleanliness of the surgery. It takes humans time to get good enough at these surgeries. Second, operation of the automated device still requires that the materials involved be generated and handled by trained personnel. Learning how to make good internal solution (which is required for the success rates reported here) is a time-honored rite of passage in physiology labs around the world, meaning it takes training. Making patch pipettes of the correct size and resistance also requires familiarity. The list goes on, and the ultimate picture is that trained professionals who specialize in these techniques are still necessary for use of the automated machine. Overall this lessens the impact of the advance described here, since staff with the necessary training to master all of the other required elements (some of them listed above) will probably not need the use of automation of this particular step. The authors should consider and comment on this wider issue.

Response: We believe many electrophysiologists have a dream like us to fully automate in vivo single-cell labeling with electrophysiological characterizations. The current study made one important step towards this ultimate goal. As commented by the reviewer, we fully recognize the technical difficulty we may face thus even in the Discussion we didn't comment on it because we know it may still have a long way to go, assuming it can be realized. And a fully automated system covering the lab work from solution making, craniotomy performing, electrophysiological recording and electroporation is clearly beyond the scope of the current study. But the good news is we are not alone. To our knowledge, several labs and a couple of companies have been working on automating small animal surgeries especially the craniotomy. On the other hand, we also know several prototypes of more affordable, miniature headstages have been developed for in vivo electrophysiology. Based on our own expertise we thus think it would be better for us to focus on what we are good at, and have collaboration with these groups if possible. But the fast progresses demonstrate the need for automated electrophysiology and we are glad to see it happen.

Minor issue #1) In Fig. 3G and 4C, comparing the "success rates" of the automated and manual approaches, it would be helpful to label these directly on the figure (rather than obliging the reader to visit the figure legend to figure out what is going on).

Response: We thank the suggestion by the reviewer. We have updated the figures accordingly.

Itemized Response to Reviewer 2:

Major issue # 1) Neuron identification/sealing: (A) The system identifies neurons and obtains seals in a manner very like that described in earlier work by some of these authors (Kodandaramaiah et al., Nat. Methods 2012). Yet there are subtle differences that aren't noted or explained. (1) Kodandaramaiah et al. use 800-1000 mbar when pipettes enter the brain. Here the corresponding pressure is only 200 mbar. Why? Does the higher pressure damage tissue or eject too much intracellular solution into the extracellular space?

Response: We want to clarify that experiments conducted in the present study are quite different from those in Kodandaramaiah et al., Nat. Methods 2012, e.g. in the Autopatcher paper mice never survived for 4-7 days to express extraneous proteins, thus different methods are required. For reviewer's questions, there are two main reasons, namely the pipette tip size and animal preparation, accounting for the difference of high-positive pressure between the current study and Ed Boyden's work. Firstly in the current study we used larger tip-size pipettes (3 - 6 M Ω), compared with patching pipettes (3 - 9 M Ω) used in Kodandaramaiah et al., Nat. Methods 2012; Secondly we made small openings in dura to facilitate the penetration of the pipette thus a very high positive pressure wouldn't be necessary. The choice of the high-positive pressure value was supported by the experiment data: under our conditions

the rate of rejecting penetrations caused by clogged pipettes was low ($10 \pm 2\%$) and comparable to published manual and automated patch-clamp work in vivo (Kodandaramaiah et al., 2012; Desai et al 2015). Lastly, we haven't directly compare the tissue damage when using different high-positive pressures so we don't have a definitive answer, but in general as the reviewer pointed out, using lower positive pressure would eject less K-gluconate based internal solution, which would help to reduce possible physically- and/or chemically-induced brain tissue damage.

(2) Kodandaramaiah et al. describe a multi-step process of detecting a neuron (with an absolute criterion that depends on the initial pipette resistance). Here the procedure seems to be simply an 8-10% increase. Again, why?

Response: We would like to clarify that as we mentioned in our manuscript, absolute and relative detection threshold were both used for manual whole-cell and cell-attached recordings for a long time (e.g. Magurie et al 2002; Desai et al 2015), thus our detection criteria are quite normal. And as we have explained in our response to reviewer#1's comment **2.a**, this could better accommodate the variations in pipette resistance (please also see page #4 paragraph 3 of the main text) thus reduce the rate of possible false negatives (misses). Our data showed it represented a good compromise between efficiency and accuracy of detection.

(B) What does the system do if the detection verification test fails (no spontaneous or evoked spikes)? Does it withdraw the pipette or does it keep moving to find another neuron?

Response: This is correct. Those penetrations were labeled as 'non-neuronal detections' and the pipette would be retracted to the surface of the brain. We have described this in our **Supplementary Information** (page# 3, paragraph 1): "Failure of evoking spikes after current injections mainly suggested false positives, which could be glial cells, blood vessels or clogged pipettes".

(C) Can the pipettes be reused since they're not forming tight seals? That is, can one record/electroporate a neuron, detach, move laterally, and search for another neuron?

Response: The same pipette can be reused for multiple penetrations if it was not clogged. The typical operation is once a penetration was decided to terminate, high positive pressure would be applied to the pipette before the pipette retraction. The pipette would then be rapidly retracted to the surface of the brain. If dye efflux was observed, the pipette can be moved to a new location and reused for the next penetration. We want to clarify that we never moved and strongly recommend not to move the pipette laterally when it is still in the brain. We added this piece of information to the revised manuscript.

Major issue # 2) R_p+R_{cleft} : How much does the value of the total resistance (R_p+R_{cleft}) affect the quality of the cell-attached recording? The manuscript is rather vague about how spikes are detected. From the figures (for example, Fig. 2), the identification does not appear to be ambiguous, but the authors should describe precisely what it was (absolute threshold, threshold in standard deviations, maximum-minimum spike width, canonical spike shape, etc.). They should also test whether spike detection depended on R_p+R_{cleft} . Without that test, it's hard to know what Suppl. Fig. 2 is showing us.

Response: We appreciate the suggestion and have added annotations to **Fig. 2**. In our dataset we didn't notice apparent correlation between R_p+R_{cleft} and quality of electrophysiological recording, because we have tried to optimize our parameter set to avoid too high or too low R_p+R_{cleft} values (see **Suppl Fig. 3**). R_p+R_{cleft} had a similar distribution between the neuronal detection events in which we recorded spikes either spontaneously or evoked by current injections and those 10% non-neuronal detections where detection was triggered but no spikes could be seen. For spike detection, we assume the

reviewer meant detection spikes after current injection. This was done as we wrote in the **Current injection** section of the Supplementary information: “The signal within 1 sec post current injection was thresholded at 3 - 5 standard deviations to detect evoked spikes”.

Major issue # 3.1) SCE: A single procedure (12 V, 1-2 ms pulses at 50Hz for 0.3-2.0 s) was used for electroporation. Was this procedure ever varied to determine whether it was indeed optimal? And did the success rate of SCE depend on the quality of the seal ($R_p + R_{left}$)? (This second question is the one that I think is most important and that many researchers would like to know.)

Response: In the present study we did vary the voltage, as we stated in the main text (page# 6, paragraph 3): “To prevent post-electroporation cytolysis, electroporation voltage was also lowered so that the trans-membrane current is approximately between 1 – 1.5 μ A, which is estimated to achieve the dielectric voltage drop across the cytoplasmic membrane”. We think this is important for successful electroporation. For the second question about whether SCE correlates with the seal, we also looked at this issue. Briefly as we stated in our manuscript, high R_p would increase the chance of clogged pipette, and too high $R_p + R_{left}$ may cause detachment problem. Within the range of $R_p + R_{left}$ we achieved, we didn’t found any apparent correlation between the value of ($R_p + R_{left}$), or R_{left} alone or relative R_p increase and the chance of successful electroporation. We will continue working on the search for the indicator of successful electroporation. For now we think more work is needed to further look into this issue for better insights.

Major issue # 3.2) Multiple penetrations: The value of automating these processes, as the authors note, is the potential to maintain high throughput. One thing that limits high throughput is tissue damage because of multiple penetrations. Do the authors have any data on this question? Their visual cortical craniotomies are quite large (4 mm diameter), suggesting that by spacing electrode penetrations the authors could use many, many electrodes per craniotomy. Is this right? Put another way, what limits the number of neurons and neuronal locations that can be obtained with this system?

Response: On average 9 – 15 penetrations were conducted per animal (main text page#4, paragraph 3). We have examined this tissue carefully but didn’t notice apparent signs for tissue damage (**new Suppl. Fig. 10**, please also see our response to reviewer#1’s comment 4). For us two known limits are: 1) labeling density that processes from different labeled neurons can be reliably separated; 2) time available for experiment as our animal protocol implements a maximal length of time for holding animal under anesthesia.

Major issue # 4) Automation: The authors should describe more clearly what parts of their system are automated and what parts are not. They note that the current injection / neuron identification part includes some leeway for the experimenter to intervene, but it’s not clear what they mean. (By the way: “man-in-loop” should be “human-in-the-loop”. The second term is what appears in the engineering literature nowadays, and so is the preferred term of art.) From the WMV file (1932465), it appears that interaction with the Multiclamp Commander is not automated. Why not? It should be since the Commander is a piece of software.

Response: We have changed the term “man-in-loop” to “human-in-loop”, and clarified the automation in the revised manuscript. Although current injections (interaction with the Multiclamp Commander the reviewer referred to) can be automatically done, they were intentionally left manual, the rationale for this was explained in the **Current injection** section of the Supplementary Information. A more autonomous system can be made, in fact the entire penetration could be automated but we intentionally kept the human confirmation or override options as from our own experience we felt some flexibility would be important for these highly complicated experiments. A scenario we had before was,

when we started the experiment we planned not to use current injection but just after a couple of penetrations we found it necessary. The current design allows the user to switch freely. We recognize a fully automated system is attractive, but from our limited experience some human intervention may still be needed for such experiments in living brain. What we can do perhaps is to provide a version of the software package in which the “full auto” mode can be enabled by the user, if it will better address the reviewer’s concern.

Major issue # 5) Equipment: How much flexibility is there in the choice of equipment? For example, say that a researcher had a manipulator from Luigs-Neumann? How difficult would it be for them to adapt the authors’ system and how quite would they do it? How about people who have DAQ devices from National Instruments or HEKA? Is Digidata required for some reason? And what about the Neuromatic control box? The authors write that in their earlier paper the functions of the control box were custom made but that they are now available in commercial form through Neuromatic Devices. That’s fine, but what exactly are those functions? Those readers who have read earlier work from the Forest group will know, but this publication – which, like all publications, should be as standalone as possible – should spell it out. A related issue: how expensive is the Neuromatic control box? For a paper of this sort to have value, it must persuade other researchers to try out the methods. A big stumbling block is money. As gauche as it may be to point this out: if the box costs \$1000, many people will be tempted to give it a go; if it costs \$10,000, many of those same people must stop and think hard; and if it costs \$100,000, most of the remainder will immediately stop reading the paper. The Neuromatic Devices website provides no guidance on this point – only a link to communicate with (presumably) sales staff. The alternative is a custom-made device. How hard and expensive would that be?

Response: In the design phase equipment independence was considered. Although it’s difficult for us to implement a fully equipment independent platform since we are not professional software developers, due to its design ACE doesn’t heavily depend on certain types of equipment. For example, ACE has no requirement for specific brand names of the micromanipulator or current source at all so a Luigs-Neumann manipulator can be readily used. Digidata was chosen because it appears to us that many electrophysiology labs including us have one or more Multi-Clamp systems. But ACE is largely independent of data acquisition hardware or software: in the current design it just needs to tell the data acquisition device when to switch between current- and voltage-clamp, and that’s it. We appreciate the diversity of equipment among labs, as we know some labs prefer commercially available software/hardware packages but others develop their own fully-customized software. This imposes challenges on developing a universal platform but it is our plan to receive more feedback from the community, to evaluate whether resource should be allocated to develop a more autonomous platform.

The Neuromatic control box is recommended since it works best with the control software. As for the price, we got the Neuromatic control box with a deep discount, since we are developing this new robot. We suggest the reviewer contact Neuromatic Device directly to get more pricing information. But interested users may build their own control box following instructions listed in a recent paper Kodandaramaiah et al., Nat. Proct 2016, and the cost will be under \$2,000.

Minor issue #1) In Fig. 1A, the Control Box also provides pressure to the pressure port of the electrode holder. This should be indicated in some manner.

Response: We thank the suggestion. We have updated Fig. 1 accordingly.

Minor issue #2) In Fig. 2A2, there is an initial drop in R_p from 10.2 to 9.4 MOhms. Is that simply a consequence of changing the positive pressure from 200 mbar to 20 mbar? If so, that should be noted in the figure caption.

Response: We don't think the initial drop of Rp is a result from pressure changing since it didn't occur every time. The data suggest the pipette passed something in the brain, most likely a dendrite or capillary so it represents a rejection. We added this to the figure caption.

Minor issue #3) The AVI file (1932466) in the supplementary material won't play. I tried it on multiple computers but always got an error.

Response: We apologize for the inconvenience. We have updated the movie file.

Itemized Response to Reviewer 3:

Major issue # 1) What is not clear are the novel aspects of their approach given that a number of different groups have already established the majority of the methods described? As described, the data appear as incremental steps in the optimization of existing methodologies rather than a completely novel approach.

Response: We think the novelty lies in 1) ACE is the first robotic platform reported linking structure and function at the single-neuron level, which is a fundamental question in neuroscience; 2) it delivers improved efficiency with standardized experiment procedure, which greatly reduces the variability of experiments and dependence of individual experimenters; 3) it lowers the entrance bar of important but complicate *in vivo* experiments so more labs can join the party. We consider it novel because in our literature search we haven't seen any published automated platform that can do what we have presented here. It is possible that similar robot was already made before but to our limited knowledge we just don't know. In this case we must declare ignorance and we will be glad if the reviewer can correct us. Since we have the same goal of conducting "a completely novel approach", we appreciate the high hope the reviewer had on us and we will continue to work hard to meet this hope.

Major issue # 2) The first rather misleading aspect of the manuscript is the title. As read, it suggests that the authors use novel technologies to perturb some aspect of neuronal activity in order to better understand function. However, the authors do not show data where cell function has been manipulated, instead they express EGFP, which is not a perturbation, rather a knock-in with hopefully no physiological effect on function.

Response: We agree with the reviewer that the title may be vague. For better clarity we have changed it to "A robot for high-yield, *in vivo* single-neuron electrophysiology and full morphology". By "perturbation" we meant manipulation of genetic/chemical contents of single brain cells. We thought it had a broad coverage because expressing actuators such as Chr2 to alter neuronal activity (the "all optic approach" we mentioned) was actually done through changing genetic information within the cell to express these proteins. In the current study expressing GFP just served as an indicator for successful manipulations.

Major issue # 3) The authors then suggest that the technical advance provided by their robotic Automated single-Cell Experimenter (ACE) and Single-Cell Electroporation (SCE) is a significantly higher yield than previously published methods. But the authors compare their statistics with a publication from the Boyden lab that focused on attaining whole-cell patch-clamp recordings, where it is likely that 'failed' patches were recorded in the cell-attached configuration (Kodandaramaiah et al., 2012). Also, the approach used negative suction to attain >1GOhm seals which is very different to the 'atmospheric pressure' release approach used in the present study. This is a much simpler method to attain cell-attached recordings, which presumably can be achieved using any of the existing 'autopatch' methods. The authors then compare the efficiency of their method with a paper that focused on patching in

awake, behaving mice (Desai et al., 2015). The brain displacements that occur in awake mice significantly reduce the possibility of attaining cell-attached or patch recording configurations, so it is rather odd to compare recording efficiency in two completely different preparations. Therefore, I don't really see this as a significant technical advance, rather an optimization of existing technologies.

Response: To clarify we measured the success rate for each key process and compared results with corresponding published works (please also see our response to reviewer#1's comments **2** and **5**). For instance, Kodandaramaiah et al., 2012 and Desai et al., 2015 were cited for neuron detection. But in terms of electroporation, Oyama et al 2013, Cohen et al 2013 and Dempsey et al 2015 were cited.

For the comparison between ACE and Autopatcher, in the present study no negative pressure was ever applied, as we stated in our manuscript (**Supplementary Information** page# 3, paragraph 1). This is because once forming a Giga-seal, an "outside-out" patch needs to be made in order to safely detach the pipette from the cell, which we found quite difficult to achieve. That's why an 'atmospheric pressure' release approach was used here. As users of the Autopatcher, we think a main goal of Autopatcher is to reduce the rate of 'failed' patches because it was designed to perform whole-cell patch recordings automatically. Thus it will be great if the reviewer could share with us on how Autopatcher would efficiently conduct the work described here. And from our experience we don't think failed whole-cell recording attempts will give good cell-attached recordings. Typically in such failed attempts the membrane will get ruptured and the data won't be reliable. This is exactly what we want to avoid.

Major issue # 4) The additional 'novel' features of the robotic system are the 1) automated AP generation during cell-attached configuration to enhance cell detection and 2) automated electroporation of exogenous substances. Unfortunately, both of the approaches can be simply automated via existing software packages when performing manual 'blind-patch' recordings in vivo. So again it is not clear what advantage is gained from implementing these technologies. Moreover, the ability to deliver plasmids and express EGFP via in vivo electroporation has been shown before (Kitamura et al., 2009; Rancz et al., 2011) so again the data appears to be a subtle optimization of existing technologies.

Response: As the reviewer commented, two-photon guided single-cell electroporation has been done previously (e.g. Judkewitz et al. 2009; Kitamura et al., 2009) but restricted in superficial layers. Our platform can work in deep brain structures so projection neurons can be labeled and traced, as we had shown in our manuscript. L5 and 6 pyramidal cells provide the long-range projection, which is a less-charted area in our opinion. Rancz et al., 2011 used whole-cell recording to dialyze DNA plasmid into neurons and their results are truly impressive. But their approach was manual and quite difficult to be adopted by other labs especially those without too much experience in electrophysiology. In vivo whole-cell recording by itself is challenging, not mentioning their approach will require careful withdrawal of the pipette to form an "outside-out" patch to reseal the membrane so the neuron can survive. We consider ourselves relatively experienced in in vivo whole-cell recordings but so far we haven't had any major advance with this technique.

Major issue # 5) The authors state that intracellular infusion of biocytin during patch recordings – seen as the gold standard technique for morphological reconstructions in vivo – may not capture the full morphological extent of a neuron. What evidence is there to support this claim? To what extent is EGFP expression more efficient – this should be shown as experimental data. The authors then seem to contradict themselves by stating "It has been shown that cell-labeling with fluorescent proteins is comparable to biocytin-cell filling". I agree with the authors that single cell electroporation spatially restricts protein expression but the number of 'ghost cells' when using biocytin can be significantly reduced or controlled by reducing the total number of penetrations per preparation. In the paper the

authors perform 9-15 penetrations to infragranular layers – given the standard geometry of patch pipettes, this high number of penetrations will inevitably damage the cortex so it is not surprising to find that many non-recorded ‘ghost’ cells display biocytin staining. Simply reducing the total number of penetrations will significantly reduce the number of ‘ghost cells’.

Response: We have conducted brain-wide reconstruction of some labeled neurons and results from one neuron were shown here (see the **Figure for Reviewers**). In our opinion our labeling gave stronger signal (as GFP was continuously synthesized for 4 – 7 days) and clearer image of the soma and proximal processes (as no contamination, see Figs. 3 and 4, Suppl. Figs. 5 - 10). We have no doubt that biocytin labeling will give excellent results by real experts but no every lab can acquire such expertise (clearly not us). Recording sub-threshold activity with whole-cell recording and infusing the cell with biocytin/DNA plasmid will be superb, but it is technically much more challenging because it requires careful withdrawal of the pipette to form an “outside-out” patch to reseal the membrane so the neuron can survive, which we haven’t got any algorithms to do it reliably. For possible tissue damage, please see our response to reviewer#1’s comment **4** and reviewer#2’s comment **3.2** (also Suppl. Fig. 10). The point we want to make here is: with comparable number of penetrations, less contamination was observed in experiments conducted with ACE.

Figure for reviewers. ACE reveals fine detail of proximal neurites and long-range projecting axons. A. Montage of Z-projection images of confocal image stacks of a labeled V1 L5 projection neuron electroporated by ACE with EGFP plasmid *in vivo*. Note that little or no damage was observed after electroporation and 7-day expression of GFP. **A1.** Close-up view of the boxed region in **A**. Note those intact L1-innervating neurites. **(B)** Initial reconstruction results of this neuron labeled. Note those long-range, subcortically projecting axons.

REVIEWERS' COMMENTS:

Reviewer #1 (Remarks to the Author):

This revision is a major improvement and the authors have addressed my concerns.

Specifically:

There were several points where more clarity was needed in quantifying the success of the method. They provided these clarifications.

There were some pieces of missing information, and these were added.

There were some language issues and overstatements relative to the existing literature, and they have somewhat toned down these points.

The new title is an improvement; though perhaps the authors could drop "and full morphology" since the robot does not provide this - it requires subsequent histology.

I am happy to recommend publication.

Reviewer #2 (Remarks to the Author):

The authors have addressed my comments satisfactorily. I have no new ones.

Reviewer #3 (Remarks to the Author):

The manuscript by Li and colleagues describes a robotic platform that aims to enhance the efficiency of performing automated single cell electrophysiology and electroporation experiments in anesthetized mice. The authors provide evidence to show that implementation of this technical platform will facilitate investigation of neuronal structure-function relationships and cell-type classification in vivo. In general the authors have addressed my previous concerns and have gone some way to highlighting the advance that a system such as this would provide. There are still some issues regarding novelty (e.g. the extent to which software driven electroporation is a technical advance) but in general the ACE/SCE robot will be of interest to the neuroscience community and beyond. There are still a couple of comments that require to be addressed.

Major issue #3

Perhaps the authors misinterpreted my previous comment. I was referring to the point that in the Kodandaramaiah et al., paper the success rate for achieving gigaseal formation ($>1\text{G}\Omega$) was 36% and whole-cell configuration 32.9%. A presumed combined success

rate of ~70% (?). The authors compare this value to the 90% success rate for cell-attached neuron detection. But isn't this comparing apples and oranges given that the success rate comparison is between gigaseal formation and the 'easier to attain' cell-attached configuration? I think that the data the authors present is convincing but when they state on L132-133 that "our detection algorithm ACE achieved a better detection and recording efficiency of single brain neurons in vivo compared with previously published studies" and cite the Kodandaramaiah et al., 2012, this can be a little misleading. Perhaps they should just state how efficient their system is in achieving cell-attached recordings in vivo without the direct, in my view rather unfair, comparisons.

Major issue #5

I think this is a fair claim and it is supported by the data but perhaps the wording should be changed to be less negative about biocytin labeling as this does work well for a number of groups. The authors should also include this important control data in the main manuscript.

Itemized Response to Reviewer 1:

Major issue # 1) The new title is an improvement; though perhaps the authors could drop "and full morphology" since the robot does not provide this - it requires subsequent histology.

Response: We changed the title to "A robot for high yield electrophysiology and morphology of single neurons *in vivo*", we hope this would help to address the reviewer's concern.

Itemized Response to Reviewer 3:

Major issue #3) Perhaps the authors misinterpreted my previous comment. I was referring to the point that in the Kodandaramaiah et al., paper the success rate for achieving gigaseal formation (>1GOhm) was 36% and whole-cell configuration 32.9%. A presumed combined success rate of ~70% (?). The authors compare this value to the 90% success rate for cell-attached neuron detection. But isn't this comparing apples and oranges given that the success rate comparison is between gigaseal formation and the 'easier to attain' cell-attached configuration? I think that the data the authors present is convincing but when they state on L132-133 that "our detection algorithm ACE achieved a better detection and recording efficiency of single brain neurons *in vivo* compared with previously published studies" and cite the Kodandaramaiah et al., 2012, this can be a little misleading. Perhaps they should just state how efficient their system is in achieving cell-attached recordings *in vivo* without the direct, in my view rather unfair, comparisons.

Response: We agree with the reviewer and have removed this comparison from our manuscript.

Major issue # 5) I think this is a fair claim and it is supported by the data but perhaps the wording should be changed to be less negative about biocytin labeling as this does work well for a number of groups. The authors should also include this important control data in the main manuscript.

Response: We have revised the Introduction to give a more positive view of the biocytin technique. As we stated in our manuscript, we totally agree that it is the gold standard technique for labeling cells *in vivo*. As for the inclusion of the reconstructed cell, as the Reviewer#1 pointed out, full morphology reconstruction is not the major goal of this manuscript, which we agree. So it appears to us a possibly better idea to include this cell, together with other cells in a different manuscript which focuses on single-neuron morphology. We hope we can have your understanding here.